# Perception of the purchase budget (BGT) and purchase intention in smartphone selling industry: A cross-country analysis

**Karamoko N'da** [1]* , **Jiaoju Ge** [1] , **Steven Ji-Fan Ren** [1] , **Jia Wang** [2]

**1** School of Economics and Management, Harbin Institute of Technology, Shenzhen, China, **2** School of E-Commerce and Logistics, Suzhou Institute of Trade & Commerce, Suzhou, China

☯ These authors contributed equally to this work.
* andabuy@yahoo.com

## Abstract

The study explores the direct and mediated impacts of customers' perception of purchase budget (BGT) on purchase intention (PIT) through perceived quality (PPQ), perceived price (PPR), and perceived benefit (PB) in a cross-country setting to understand BGT's role in predicting customer purchase intention in smartphone selling through international online shopping platforms. An online survey was conducted in Kenya, France, and the United States to gather data from 429 consumers who had recently purchased one or more smartphones through international online shopping platforms. SmartPLS-4 was used to test the hypotheses. Results for the entire sample showed a significantly positive mediating role of PPR and PPQ between BGT and PIT. However, the mediating roles of PPQ and PB were not significant in the samples from Kenya, France, and the United States. The results also showed that PPR plays a significant and positive mediating role between BGT and PIT in samples from Kenya, France, the United States, and overall. However, the direct relationships between BGT and PPQ, PPR, and PB are shown to be negatively significant.

## 1. Introduction

In this digital era, smartphones have unquestionably become essential tools in people's personal and professional lives. Smartphones are almost indispensable for efficiently accomplishing tasks, such as emailing, managing messages, images, videos, geolocation, or performing online purchases [1]. Therefore, smartphone demands on the international market have been growing steadily over the past decade [2]. The number of people with smartphones is expected to reach 5.9 billion in 2025 [3]. In developed nations such as the United States and France, the penetration rate of smartphones has been steadily increasing over the past years, reaching 298 and 50 million smartphone users, respectively, as of 2022 [4]. That rising tendency is also seen in developing nations, such as Kenya, where smartphone users reached over 26.0 million in September 2021 [5]. For that reason, smartphone demand has become a topic of both practical and theoretical importance [1]. Sellers and smartphone brands are worried about knowing the best strategy to put in place to attract more customers and accomplish their sales goals in the

**Data Availability Statement:** Data are within the Supporting Information files.

**Funding:** This study, financially sustained by the National Natural Science Foundation of China (Project No. 71402039 and Project No. 71831005),

Shenzhen Peacock Program (Project No. KOCX2015032715503970). The funders had no role in study design, data collection and analysis, decision to publish, or preparation of the manuscript.

**Competing interests:** The authors have declared that no competing interests exist.

context of growing competition. Because they still do not know until now which variables may really impact smartphone customers' purchase intentions to modify customers' buying decisions when the purchase is carried out in an online shopping environment [6]. Especially on international selling platforms where e-sellers and smartphone brands deal with customers from disparate economic contexts. To answer this worry, some researchers proposed that e-sellers and brands include customer purchase budgets in their selling strategies [7]. These scholars think it could be a great strategy to attract more customers and accomplish sales goals while meeting customer needs [8]. However, these scholars did not precise what aspect of customer purchase budget should be considered. Accordingly, in order to advance the research and understand the underpinnings of the customer's purchasing budget, this study focused on the customer's perception of the purchasing budget.

A customer's perception of purchase budget (BGT) incorporates both the perception of his planning obligations and his choice expectations [9, 10]. BGT is a concept that refers to a set of processes by which a buyer becomes conscious of and interprets his spending plan in relation to purchasing costs [10, 11]. This awareness and interpretation of the expenditure plan influence the way the actor selects and responds to purchase proposals while analyzing the present and future costs as well as the advantages associated with these proposals according to their requests [11, 12]. On international selling online platforms where e-sellers deal with customers from disparate economic contexts, considering BGT may be a relevant strategy for e-sellers [13, 14]. E-sellers can expect positive customer responses by offering products that match the customer's perception of the purchase budget (BGT). Similar to this, with increasingly aggressive competition in the smartphone-selling industry, selling more products and making more profit have always been crucial goals for e-sellers and brands [1]. Thus, considering BGT in the selling strategy can help achieve it and attract customer intention, which is a key illustration of the degree to which a customer would like to buy the product [15]. Even though research has suggested exploring the relationship between BGT and customer purchase intention (PIT) in the smartphone-selling industry, almost no studies until now have focused on it [7]. Therefore, whether BGT significantly explains customer purchase intention in the smartphone-selling industry is still not empirically resolved [7]. As a result, this issue requires to be examined.

Customer purchase intention is agreed to be a complex behavioural foundation [16]. Thus, researchers that have proposed to study the link between BGT and PIT suggest the inclusion of additional variables with explanatory strength, such as perceived product quality (PPQ) [7], perceived price (PPR) [17, 18], and perceived benefit (PB) [18]. These researchers pointed out that these variables could impact customer purchase intention directly and significantly [19–21]. Similarly, the mediating role of these variables in the connection between BGT and PIT has not been explored until now in the smartphone-selling industry. Therefore, the process by which BGT affects PIT has not been clearly established yet. Even if the existing research has suggested that BGT could be, in fact, a predicting factor of customer purchase intention, they have not specified the direction of the link, i.e., direct or indirect. Moreover, research on BGT's indirect or direct relationship with PIT in the smartphone-selling industry is scarce. Additionally, even though the linkage between BGT and PIT has been suggested to be examined and appears obvious, it is still unknown if buyer perceptions and answers towards BGT are the same across different settings or countries. Although examining consumer diversity is essential, there is disagreement about whether and to what extent country-setting variations should be considered [16].

Although more smartphone e-sellers sell to various customers on global online selling platforms, adopting a great selling strategy based on the budget may be more difficult on that scale because of country-setting variations. The customer responses and perceptions of the purchase budget (BGT) might not be the same everywhere. Countries' purchasing power, tastes, and

living conditions might generate differences [22, 23]. Therefore, considering customer differences becomes crucial for e-sellers looking to succeed in the global online market [24]. In this context, the tiers of countries' economies may be utilized to explain the notable variances in buying power and living standards worldwide [25]. Regularly, consumers in developed nations show tendencies to purchase more expensive products than those in developing countries [26]. Customer answers might also differ based on product features [27]. Therefore, the perception of BGT could differ depending on customer preferences and quality expectations, which are frequently mirrored in their judgment process [7].

To summarize, this study mainly intends to answer the following questions:

RQ1: What is the effect of BGT on PIT in smartphones-buying on international online shopping platforms?

RQ2: What is the mediating role of PPQ, PPR, and PB in the relationship between BGT and PIT?

RQ3: Do the impact of the customers' perceived purchase budget on PPQ, PPR, and PB are the same across different settings or countries?

The current research measures and evaluates buyer perceptions in three different nations. That is to say, Kenya, France, and the United States represent dissimilar economic settings. In doing so, we expect to be able to generalize our conclusions in this way. This research seeks to close two main limitations by theoretically putting forward and assessing through observation a conceptual model incorporating PPQ, PPR, and PB as mediators between buyer perceptions of BGT and PIT. These mediated relationships are assessed within a cross-national setting. As a result, the theory of planned behavior (TPB) is used as a theoretical lens to generate the study hypotheses.

The study makes numerous improvements to earlier studies by filling up these gaps. First, considering the dearth of studies currently available on the BGT-PIT link, this research replies to the calls from previous studies to assess the link between BGT and PIT as well as mediating mechanisms through which BGT impacts PIT in the smartphone-selling industry. In this respect, the inherent relevance of the BGT-PIT relationship is barely studied. However, it is essential to study that link and its potential benefits for online smartphone vendors and brands [7]. There is a serious dearth of literature in the smartphone sales industry combining the mediators found in this study into a single model to account for how consumer perceptions of BGT affect customer purchase intentions. As a result, there is still a dearth of understanding of the BGT variable's various directions [7]. This study makes a major contribution by examining variables that could assist in describing the process used to develop a customer's buying intention using BGT. Additionally, samples from different nations are also helpful in showing the results of consumer perceptions of BGT in various national settings. The research will advance knowledge of BGT-PIT links in cross-national contexts since minimal studies have investigated that relationship. Especially in such contexts. Additionally, almost no previous study has compared consumer perceptions of purchase budget, the suggested mediators, and PIT across various country settings. The current study's cross-country context also sets it apart from earlier studies on smartphone purchase intentions.

## 2. Literature review and hypotheses development

The marketing literature has extensively discussed the budget concept [10]. However, its study is often focused on the budget constraint angle rather than its influence on the transactions. [12, 28, 29] defined it as the formal expression of plans and objectives of financial expenses expressed in quantitative and monetary terms, assisting efficiently in achieving organizational and company goals. Since then, the budget concept has evolved from expense planning to a strategic plan for efficient decision-making [12]. Therefore, in the marketing literature, the

budget is understood as a sum allocated from revenue ready to be utilized to purchase a product or service [7]. In the smartphone purchase setting, [8] defines the budget as a financial plan enabling consumers to evaluate alternative offers according to customer needs and product affordability. The budget thus imposes choice and expense obligations according to the product's affordability.

Budget research is dominated by two main theories, namely, the game theory [15, 30] and the theory of planned behavior (TPB) [29]. According to game theory, decision-makers are rational and will strive to maximize their payoffs such that their actions affect their choices [15, 30]. The TPB argues that intention is the best indicator to predict someone's future conduct and attitude. In this perspective, the prior setting of a budget allocated to purchasing a product, for instance, can be considered as the manifestation of the buyer's intention of choice. That buyer's intention is determined by certain perceived values that the buyer wishes to own through the product, e.g., quality and benefit. TPB has been used to explain various choice behaviours in several fields, i.e., tourism, service quality, and banking [29]. [8] pointed out that since consumers may be sensitive to price due to their purchasing capability, focusing on their budget plan is imperative because it allows them to evaluate alternatives and significantly affect customers' purchase intentions. That being so, in the smartphone industry, brands, sellers, and resellers need to pay attention to the customer's budget, as it can be relevant to customer purchase intention [7]. When doing so, prior literature shows that considering budgets in buying, selling, and managerial activities are associated with several positive outcomes. Among these include the positive impact of mental budgeting attitude on household purchase intentions [29]; the elucidation of the near-optimal dynamic mechanism of online auctions sales for purchasers facing a limited budget [32]; the characterization of procurement design mechanisms with a fixed budget [33]; the positive influence of budget on shopper spending [34]; and improving under limited budget companies' managerial responses strategy through optimal guidance to online reviews [35]. Other studies pointed out the negative effect of a limited budget on online auction advertisers' profit [30] and the welfare-maximizing mechanisms of a population [36]. Additionally, other researchers have reported significant impacts of budget conditions, including the significant influence of tightening budgets on optimal choices and improved budget allocation strategies in advertising markets [15, 37]. However, this paper focuses on the link between customer perception of purchase budget allocation and purchase intention, especially for smartphone purchases on international online shopping platforms.

The budget has become a potentially relevant attribute within the marketing setting. That is so because, in product markets such as smartphones, where one may always find new product brands that are increasingly sophisticated and high-quality, attributes such as product quality are not any anymore a key advantage in the market. However, prices are seen to be one of the main variables that might prompt customer responses. In these conditions, given that the product price is an attribute that online sellers offer according to their profit, potential customers may be more or less sensitive to the price [8]. This sensitivity may be stemmed from potential customers' purchasing capability. A budget is an amount of money planned to be spent on purchasing a product or service. That amount may be less or more than the product or service price and influences potential buyers' purchase intentions. Therefore, verifying the relationship between budget perception and purchase intention is imperative, as this relationship has been very little explored in the smartphone-selling industry.

## 2.1. Theoretical foundations

In the case of smartphones, a CBEC transaction's success depends on the strategies put in place by e-sellers to convert subjective perceptions of buyers on the product (e.g. perception of

quality, price, benefit, etc.) into buyer purchase intention and decision-making. The distance between buyer and seller requires e-sellers to develop strategies to convert buyers' subjective perceptions of products into purchase intentions and decisions. Also important is the alignment of the buying intention and decision-making with the perception of the budget for purchases [29, 35].

Therefore, it's critical to comprehend how a customer's purchase budget, as well as the subjective perceptions of the buyer on the product, such as the quality, price, benefits, etc., influence his purchase intention. In this regard, [29] claimed that using the theory of planned behavior, customer perceptions of their purchase budget can influence consumers' perception of the product. These perceptions, in turn, may impact the customer's buying intent [21, 29, 38, 39]. In context, it has been suggested that TPB can be used to investigate the links between customers' perception of purchase budget and customer purchase intention, as well as the mediating role of buyers' perceptions [38]. From this theory [38] concluded that customers' perception of purchase budget by influencing buyers' subjective perceptions of the product, i.e., quality, price, and benefit.., could lead to influence their purchase intention. Based on that theory lens, we argue that when customers find a match between their perception of the purchase budget and their subjective perceptions of the product they wish to purchase, this will reinforce their purchase intention, which is a result of their positive perceptions of the product. Our argument seems to be supported by [35]. They pointed out that customers' perceptions of the purchase budget could impact their perception of obtaining the values sought of the product they would like to buy, which in turn will reinforce their purchase intention.

## 2.2. Budget and purchase intention

PIT is defined as a type of behavioral intention that describes how likely a buyer is to make a future purchase of a certain product or service [40, 41]. In this perspective, the prior setting of a budget allocated to purchasing a service or product can be considered as the manifestation of the buyer's intention to purchase and choose. Identifying factors underlying customer purchase intention has been featured as one of the primary objectives of sellers and brands in maximizing business benefits and leading to business success [8]. In the smartphone market, brands and sellers aim to attract and hold customers by concentrating on the elements that are known to influence purchase decisions. In order to successfully satisfy buyers' desires, smartphone brands and sellers constantly want to be informed of the most significant and desired trends among consumers [42]. In this context, the relationship between customer purchasing capacity, as highlighted by his purchasing budget, and the desired product, should be a fruitful field in the smartphone-selling industry and for sellers striving to improve their customer purchase intention knowledge [7, 8]. The customer's perception of the purchase budget is crucial in determining the customer's choice perceptions, categorizing customer expenses and motivating its decision-making and purchase intention [8, 29]. Therefore, the customer's perception of the purchase budget should be an essential foundation in the marketing and management field [29, 43]. Firms have been the first to recognize the importance of budget setting in management [43]. Since then, the budget has become a research subject in many industries [17], including the smartphone online selling industry [8]. In this optic, [7] suggested that customers' perception of purchase budget concept should be included in future studies about smartphone purchases online to verify its effect on buyer purchase intention. Therefore, this current study somehow responds to [7, 8].

Researchers argue that the budget could significantly impact customer purchase intention [7, 8]. First, the customer's perception of the purchase budget enables the customer to evaluate the affordability of products and identify different product brand alternatives based on the

positive perceptions that the product generates [7, 29, 43]. Second, that positive evaluation of products enabled by the customer's perception of the purchase budget strengthens purchase intention. In this context, most customers believe that they are more likely to base their purchases on allocated budgets for effective and efficient decision-making, which might have a favorable budget influence on purchase intention [8, 17]. [8, 29] noted that customer perception of the purchase budget serves as a decision-making incentive and protects and controls buyer reaction to purchase proposals while examining the benefits of such proposals. Therefore, the perception of the purchase budget before the transaction might positively affect the purchase intention and customer confidence.

Buying a product depends not only on the customer's willingness to pay but also on his purchasing capability. When customers perceive a perfect suitability ratio between their purchase budget and service or product offers, that may increase their purchase intention. Especially in the online smartphone purchase framework. [7, 8] have noted that customer perception of the purchase budget might significantly impact customer purchase intention. According to previous literature, only a very limited number of studies have explored the linkage between customer perception of the purchase budget and purchase intention [7, 8, 29]. Additionally, previous literature suggests that the appropriate theory to explain that relationship is the theory of planned behavior (TPB) [29]. This theoretical framework specifies that planned behavior is determined by intention. From that perspective, one could conclude that the customer's purchase intention in terms of quality or quantity, for instance, may be driven by the perception of his purchase budget, considered as the customer's perception of his planned purchase action before the transaction. Thus, when a particular product or service matches the customer's perception of his purchase budget, that may positively increase his purchase intention. With this theory, much research in marketing has revealed positive and significant relationships between different variables [44]. For instance, [29] reported a significant connection between mental budgeting attitude and purchase intention. Similarly, [45] reported a positive correlation between perceived behavioural control and eating intention. Therefore, the findings of previous studies confirm our belief that the impact of customer perception of the purchase budget on smartphone purchase intention through international online platforms may have a positive relationship. Hence, we postulate the following hypotheses:

H1. Customer perception of the purchase budget positively affects customers' smartphone purchase intention through international online platforms.

## 2.3. Mediating effect in the budget-purchase intention link

As shown above, although topics related to the budget have triggered several bodies of research, there is still a limited understanding of various indirect and direct pathways that connect BGT to PIT in the online smartphone selling industry. The literature reviewed by [8] concluded that consumers might be sensitive to smartphone prices due to their purchasing capability. Accordingly, the possible customers first evaluate if they can pay the price of the desired smartphone before engaging in the transaction. In doing so, a customer's perception of the purchase budget could determine their purchase intention. [7, 8] suggested, therefore, verifying the impact of customer perception of his purchase budget on his purchase intention and determining which way customer perception of his purchase budget could affect his purchase intention. In other words, what factors could mediate between customer perception of the purchase budget and purchase intention in smartphone purchases? However, since that suggestion, almost no studies, to our best knowledge, have investigated that proposal in the smartphone industry. Thus, almost four years later, [7] 's proposal has not yet been investigated. The variables we have considered in this research include perceived quality, perceived

**Table 1. Budget concept study within different domains and industries.**

| Reference | source | Mediators | Budget variables | Industry | country | Results |
|---|---|---|---|---|---|---|
| [34] | Van Ittersum et al. (2013) | None | Allocated Budget | Online shopping | United States | The budget positively influences shopper spending. |
| [37] | Gendron et al. (2014) | None | Budget Allocation | Advertising Market | China | Significant influence of tightening budgets on optimal choices |
| [30] | Lu et al. (2015) | Country Income Level | Budget-Constraint | Online advertising | United States | Budget constraints negatively affect online auction advertisers' profit. |
| [31] | Habibah et al (2018) | None | Mental budgeting attitude | Household | Pakistan | The mental budgeting attitude positively impacts household budgeting intentions. |
| [33] | Anari et al. (2018) | None | Budget Feasible | Procurement Auctions | United States | The characterization of procurement design mechanisms with a fixed budget |
| [15] | Zia and Rao (2019) | None | budgeting strategies | Online advertising | United States | Budgeting strategies Improved budget allocation strategies in advertising markets |
| [32] | Balseiro et al. (2019) | None | Budget-Constraint | online Sale | United States | The elucidation of the near-optimal dynamic mechanism of online auctions sales for purchasers facing a limited budget |
| [36] | Richter M. Mechanism (2019) | None | Budget-Constraint | Economy | United Kingdom | The budget influences the welfare-maximizing mechanisms of a population. |
| [35] | Wang et al. (2020) | None | Budget-Constraint | Online reviews | China | A budget allows for improving companies' managerial responses to online reviews. |

price, perceived benefit, and purchase intention. As shown by [7, 16–18] it is evident that the different links (direct and mediating links) studied in this research are still not addressed in previous smartphone-selling literature. A summary of key studies that have examined the budget concept across various fields and industries is displayed in Table 1. As listed in Table 1, although the budget concept has been studied in some industries, it is clear that regarding the smartphone industry, not only has the budget concept not gotten much attention from scholars, but also the BGT-PIT link as well as the mediators suggested in this research have never been investigated. Therefore, this study considerably advances the earlier literature.

The link between customer perception the purchase budget and perceived smartphone quality in online sales has been neglected in the literature [7, 8]. A product's PPQ is described as the consumer perception of the value or merit of a product [46]. In the case of smartphones sold online, [7] defines the perceived quality as the apprehension of a potential buyer about the excellent condition of the smartphones put up for sale. In this perspective, [8] pointed out that the customer perception of the purchase budget is a quantitative and qualitative expression perceived of a purchase plan and an aid in evaluating and motivating decision-making. That means that customers with a clear perception of their purchase budget may better perceive the product quality they choose. In the CBEC, due to the distance between buyers and sellers and the fact that customers cannot touch or feel products, their perception of product quality might be more based on their budgets [47]. The budgets, in turn, shape their choices [48]. In this regard, [32] shows that implementing purchase budgets reduces the possibility of a buyer waiting for better purchase opportunities. Customers who believe their spending plan is in line with the proposed offer in terms of price and quality can easily identify themselves with that offer [26]. That being so, the significance of customer perception of the purchase budget in the offline or online purchase setting is fueled by the perception of the customer's demands and needs in terms of price and perceived quality [7]. Prior research indicates that the perception of the purchase budget can lead to a better reaction to purchase proposals by analyzing the future and present cost and quality benefits of such proposals [26, 29], thereby leading to improved perceived quality. In the CBEC framework, for instance, buyers come from different economic settings; therefore, considering the customer perception of the purchase budget can

result in better sales. Therefore, smartphone companies need to offer products in terms of quality that consider potential buyers' budgets [26]. Perceived price refers to a product's actual price perception level compared with the customer's reference price [49]. According to [50], the perceived price is one of the product's fundamental values and positively influences customer purchase intentions. For that reason, buyers refer to that factor in computing their purchase budget. Implementing an appropriate purchase budget and considering the perceived price generally contributes to making a good choice [50]. Because that positively affects the evaluation of the different offers [34]. Therefore, consumers tend to be more satisfied when they see that the perception they have on a purchase budget they have planned to use to purchase a given product or service matches the perceived price [7, 8, 18].

Numerous studies have used the TPB as a foundation for comprehending the PB concept. As stated by the TPB, a person's intention to adopt a specific behaviour derives from social environment factors and the evaluation of the perceived benefits [51]. PB describes buyers' perception of what they will gain from a purchase [52]. Several studies have found that consumers' perceptions that online purchases (including CBEC transactions) are beneficial are dictated by factors such as: getting more information about products, great flexibility, less effort in time, and competitive price [53, 54]. A customer with a purchase budget before purchase may easily take advantage of those factors. Therefore, his perception of the purchase budget stimulates him to track his spending and directly influences him while taking advantage of competitive prices [8, 29, 50]. The influence of customer perception of the purchase budget on customer purchase behaviour is also associated with positive customer perceptions of reducing spending [8, 7, 28]. Despite these pieces of evidence that customer perception of purchase budget might influence consumer perceived benefit when purchasing online, we found that almost no studies have explored that relationship, especially in smartphone purchases through CBEC. Hence, we formulate the following hypothesis

H2. *Customer perception of purchase budget have a positive impact on (a) perceive quality, (b) perceived price, and (c) perceived benefit*

Much research exists on the product-perceived quality-purchase intention link [46, 55–57]. However, the literature presents varied findings on that relationship. Some previous studies reported a positive relationship between these two variables. When studying psychological results on PPQ on PIT, [19] argued that there could be a positive relationship between PPQ and PIT [56], discovered a positive impact of PPQ on the PIT of certified products. In the same direction, [55] and [57] also highlighted a positive influence of PPQ on PIT. Similarly, utilizing TPB, [58] reported a positive link between products' PPQ and PIT. [46] explain that perceived product quality is an essential signal that ascertains buyers' purchase intentions. Therefore, it asserted that the fact that a customer attaches to a product brand over a competitor product is decided by its perceived quality level. That is also evident in the smartphone industry, where research showed a significant positive influence of PPQ on PIT [46, 58]. However, some studies in the smartphone and scooter or motorcycle industry reported an insignificant effect of PPQ on PIT, e.g., [7, 59, 60].

No doubt, perceived price is also represented in customer purchase intention. [19, 61] consider that perceived price acts as a mental guide that assists the customer in differentiating between products' actual price and reference price on the markets. That allows the customer to decide whether the price is reasonable or not [58]. Indeed, when customers see the product price as reasonable, they perceive more benefits in buying that product, and that increases their purchase intention [62]. Therefore, customer perceptions of price act as drivers of purchase decisions [19]. For that reason, research in marketing has been dedicated to investigating the influence of perceived price on consumers' purchase intentions [19]. Prior findings indicate that a high price tag may be a barrier to purchase probability [20]. In line with that, [19]

also found that a high perception of price negatively influenced customer purchase intention. [21] highlighted a positive connection between PPR and PIT. In the smartphone industry, PPR is also a powerful indicator of PIT [7].

In the e-shopping environment, perceived benefit is viewed as the perception of the advantages gained from a shopping action [63]. When customers feel they are profiting from a transaction, it improves their commitment to purchase action and amplifies the certainty associated with the purchase [64, 65], explains that perceived benefits reinforce consumers' trust in how much they would gain from an online purchase. Little research has focused on the perceived benefits-purchase intention link. However, the few who investigated that link highlighted a positive influence of PB on decision-making and purchase [51, 65] significantly. In this regard, [63] argued that customer purchase intention rises the more perceived benefits they receive. This has been observed in the online purchase of smartphones in the present environment of heightened competition [7]. Accordingly, it is hypothesized that PPQ, PPR, and PB may also positively affect customer purchase intention. Hence, we formulate the following hypothesis:

H3. (a) perceived quality, (b) perceived price, and (c) perceived benefit have positive effects on smartphone purchase intention through CBEC.

The literature mentioned above shows that considering the customer's perception of the purchase budget can influence the customer's perception of terms of quality (PPQ), price (PPR), and benefit (PB). Moreover, it has been shown that these customer perceptions could be crucial precursors to customers' PIT.

Additionally, it has been highlighted that the customer's perception of purchase budget (BGT) incorporates both the perception of his planning obligations and choice expectations [9, 10] i.e., his expectations in terms of price, quality, values, and benefit. Etc. TPB describes these perceived expectations as one of the foundations of customer intention [9, 10]. Moreover, TPB stipulates that customer intention is the closer determinant of customer behaviour [66]. In other words, a customer's perception of his purchase budget (BGT) could determine his expectations perception, which could influence his intention and behaviour. Therefore, a customer's perception of the purchase budget can assist in attaining positive perceptions of product quality, price, and benefits that, in turn, influence customer purchase intention. Due to the possible effects of customer's perception of purchase budget on PPQ, PPR, and PB (H2), along with the effect of PPQ, PPR, and PB on customers' PIT (H3), customer's perception of purchase budget is predicted to have favorable indirect impacts on PIT through perceived quality, perceived price, and perceived benefit. Therefore, the study proposes that consumers' PB, PPR, and PPQ mediate the effect of the customer's purchasing budget on PIT. Hence, we formulate the following hypothesis:

H4. The connection between customer purchase budget and customer purchase intention is mediated by (a) perceived quality, (b) perceived price, and (c) perceived benefit.

## 2.4. Impacts of perceived price on perceived quality and perceived benefit

The marketing literature shows that the perception of a consumer on price can influence the customer's quality perceptions and purchase probability [67]. Marketers support that idea because buyers are more inclined to think that more expensive and higher-quality goods must be priced higher [68]. Several studies have suggested that price could significantly affect the customer's perceived quality [19, 68–70]. Nevertheless, a limited number of researchers have studied the connection between PPR and PPQ, although literature suggests a possible positive and direct relationship between these two constructs [71].

Marketers strongly believe that perceived quality is heavily dependent on customer perception of price [70], which is an essential factor in perceived benefits in online shopping [63].

The research highlighted that the buyers are prepared to buy more expensive services or products if the perceived benefit can compensate for the cost incurred [71]. In this direction, [63] consider the perceived benefit as an attribute deriving from price and essential to stimulate customer purchase intention online. [72] in the same direction, showing that perceived benefit is significant in online customer shopping behaviours and affects their decision-making. Accordingly, it is hypothesized that:

H5. *Perceived price has a positive influence on customer perceived quality.*

H6. *Customer perceived price has a positive impact on perceived benefit.*

## 2.5. The link between budget and PPQ, PPR, and PB across countries

The budget concept is extensively studied in the business industry, with a large number of research concentrating on offline sales settings. However, studies on purchase budgets investigating buyer perceptions through a cross-national optic in CBEC are scarce. Far further as we are aware, almost no study contrasts the impact of consumer's perception of purchase budget on smartphone purchase intention made through CBEC markets in three different nations. As a result, cross-national research on samples from multiple nations is required to provide a more thorough understanding of the connection between consumers' perceptions of purchase budgets and buying intention. Customers from diverse economic settings could present significantly different attitudes towards the consumer's perception of purchase budget -purchase intention link. Most research has explored smartphone purchases in national and offline settings [7]. However, in the absence of certain types of smartphones on national and offline markets, potential buyers may only rely on CBEC markets.

We chose the USA, France, and Kenya for this study because:

1) These countries are significant countries in smartphone penetration rates in North America, Europe, and Africa, respectively [73–75] and 2) these countries are considered to be among the major countries in purchasing smartphones through CBEC in North America, Europe, and Africa [76, 77]. However, Kenya contrasts sharply with the USA and France regarding CBEC development levels. Nevertheless, although both USA's and France's CBEC industries are developed compared to Kenya, the USA's CBEC industry is more advanced than France's. [78, 79] indicate that B2C CBEC transactions only generated about $1000 billion in the USA in 2018, while they generated about $120 billion in France. These levels demonstrate the striking differences between the three nations' CBEC industries. Figs 1 and 2 show the conceptual model and the significant countries by smartphone penetration rate. Fig 2 has been built using data from [73–75]. We can see from Fig 2 the smartphone penetration rate in countries considered in this study and other various nations. We can observe that the countries considered in this study (USA, France, and Kenya) are significant in smartphone penetration rates in their regions, i.e. North America, Europe, and Africa, respectively. Therefore, when comparing buyers from these nations, the conclusions of our study may not be the same. Hence, we hypothesize that:

H7. *The impact of consumer perception of purchase budget on (a) perceived quality (b) perceived price (c) and perceived benefit (d) varies significantly by country (Kenya, France, and the USA).*

## 3. Research methodology

### 3.1. Survey development, data integrity and respondents' consent

We applied an SEM technique to test the study's hypotheses using SmartPLS-4. PLS-SEM modelling is a well-known method for analyzing complex business research models [80]. A descriptive, exploratory, and empirical research design was used in this study. This study aims

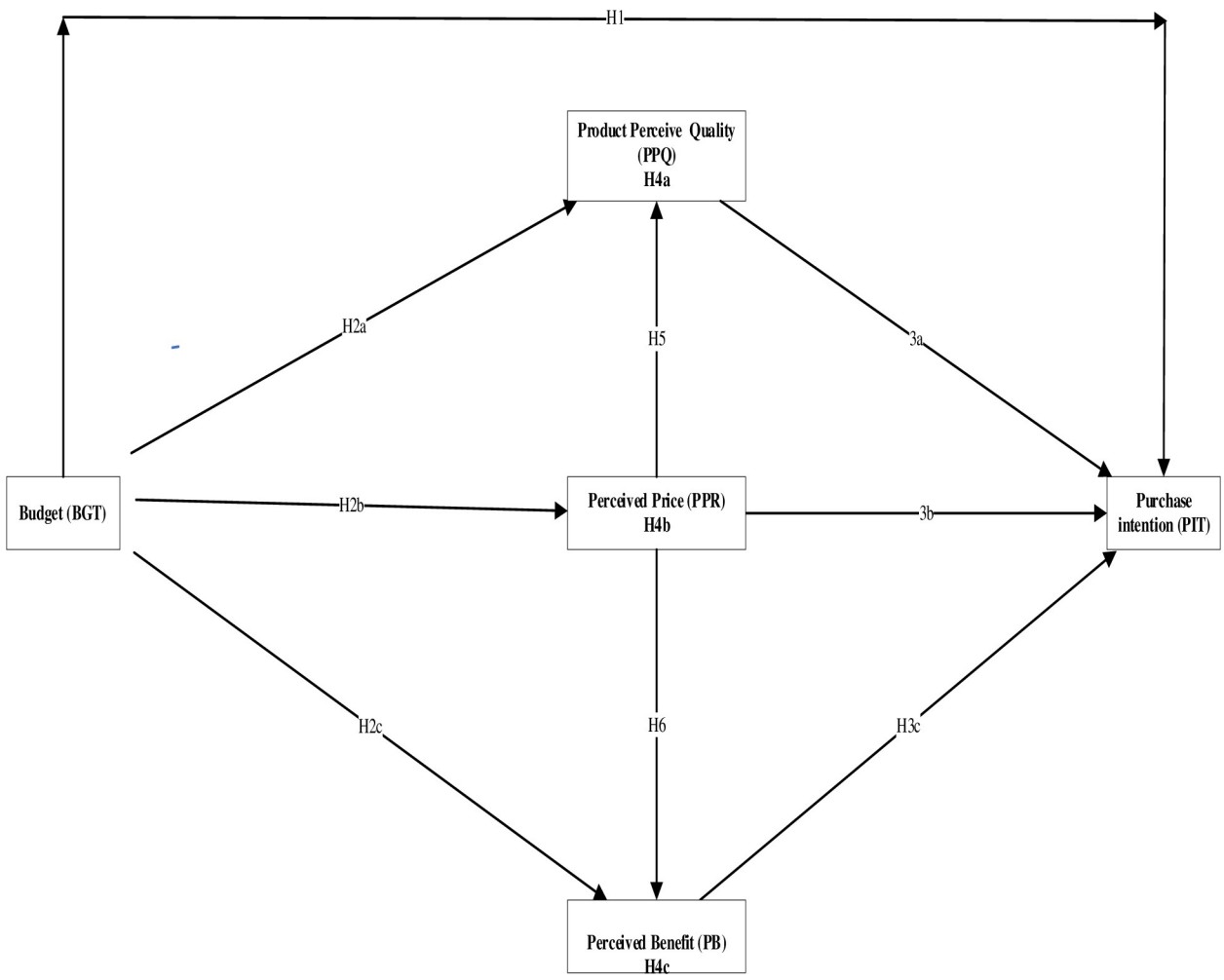

**Fig 1. Conceptual model.**

to identify the effect of customer perception of purchase budget on purchase intention through PPQ, PPR, and PB as mediators. Thus, a convenience sampling technique was chosen to carry out the study rather than a probability sample method [81]. A well-structured questionnaire was used to collect primary data for the study. The sample was collected from consumers who had recently purchased one or more smartphones through an international online shopping platform. The survey ran for about two months, from June 23rd 2022, to August 26th, 2022 through an online questionnaire distribution. We used two months to get more representative and accurate data. This duration includes data collection, and data cleaning. We created an online survey through Google Forms to collect the study sample (https://forms.gle/pS4GHXACnjNSe55i6).

Before the study began, the study received ethical approval from the committee in charge of research ethics and integrity at Harbin Institute of Technology, Shenzhen. In this direction, the research goal and survey were explained electronically to respondents. Minor customers (under 18) were not allowed to fill out the survey. All respondents gave their consent electronically before they were directed to the survey link and questions. In this regard, it was clearly written in the header of the survey that the survey was intended for academic research on smartphone purchases from CBEC platforms. Therefore, the information provided will be

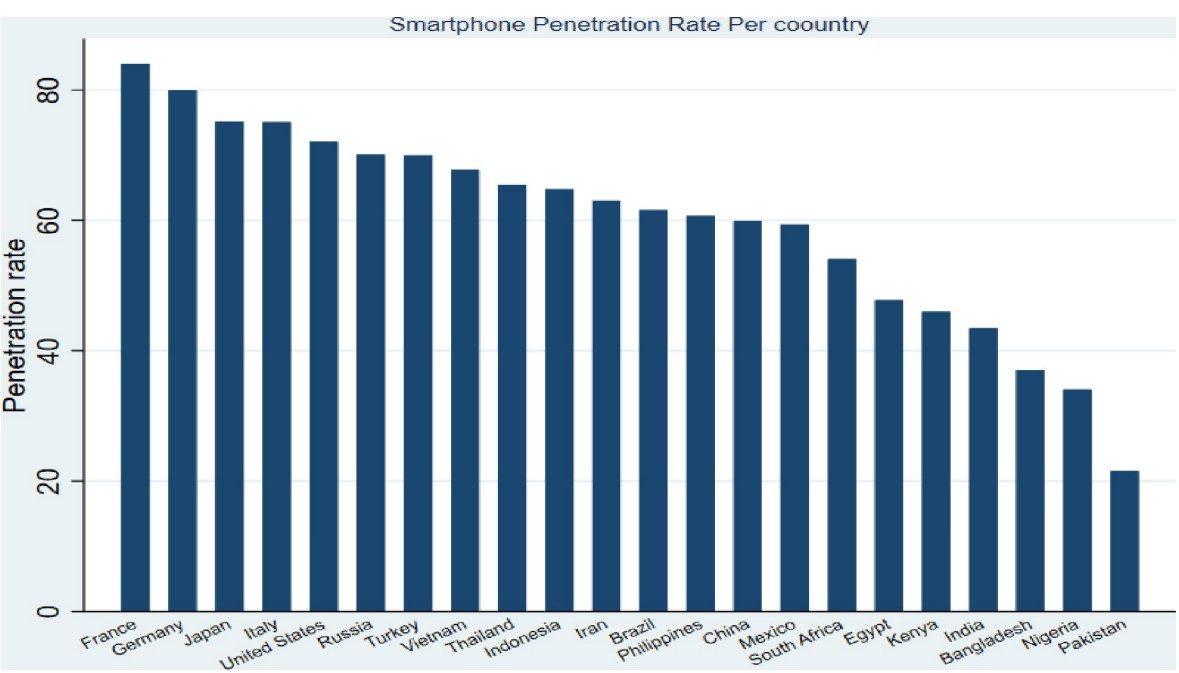

**Fig 2. Smartphone penetration rate per country.**

used for academic research purposes, including the possible publication of the results in an academic journal. The survey could be stopped by respondents at any time before it was submitted, preventing the storage of their responses. Only after respondents submit the survey at the end are their answers saved. A senior professor, head of the research data bureau in the faculty of Economics and Management at the Harbin Institute of Technology in Shenzhen, approved and validated the survey's English version first. Then, the survey's French version was translated by one of the authors of this study since his native language is French and further validated by a French language expert. We then shared the link with several online respondents (CBEC buyer groups) in France, Kenya, and the United States. The method of distributing online questionnaires was chosen because the framework of the study concerns the international online shopping market. Through this virtual framework, CBEB customers interact with e-sellers and each other, discussing and sharing their shopping experiences [82]. Therefore, the response rate would be higher as these frameworks are convenient for contacting CBEC buyers as they are the study's target respondents [82]. The respondents came from three different countries: France, Kenya, and the USA. These countries are selected for their significant commitment to CBEC sales and smartphone purchasing through CBEC platforms [78, 79].

### 3.2. Demographic questions

Demographic questions, such as gender and age, as well as the CBEC platform names where the purchase was made, and smartphone brands purchased, were asked in the first section of the survey. However, due to the missing data on the CBEC platform names and smartphone brands purchased, we did not consider them in the model.

Data collected showed that the respondents' minimum age was 20, above the minor age considered in the three countries analyzed. For instance, the minimum age in Kenya sample was 20, while the maximum was 48. In the samples of France and the USA, the maximum age

was 43 and 45, and the minimum was 20, respectively. In France, 149 responses were ultimately deemed valid. According to the sample's demographics, 78 respondents (52.34%) were men, and 71 respondents (47.66%) were women in France. The age was around 30. From Kenya groups, we collected 202 questionnaires. 62 responses were not considered because of missing data. Finally, 139 responses in total were deemed valid. Out of the 139 responders, the sample revealed that 66 (48.47%) were men and 73 (52.52%) were women. The average age was about 29 in the USA. 37 responses were not considered because of missing data. Finally, 142 questionnaires out of 178 in total were deemed to be valid. According to the USA sample, out of 142 respondents, 77 (53.91%) were women, and 65 (46.09%) were men. The US and Kenyan groups received the survey's English version. However, we sent the French survey version to the French groups.

### 3.3. Survey questions

The second section had questions to gauge how consumers perceived purchase budget, product quality, price, benefit, and purchase intention. A 5-point Likert scale was used to evaluate the items, with 1 denoting strong disagreement and 5 denoting strong agreement. Based on the definitions of [7, 28, 48, 83], eight items were used to gauge how much the purchase budget was perceived. The scale used to represent the perception of the purchase budget was two-dimensional. Based on the research of [84], four items were used to measure perceived quality. Four items that were based on the work of [85] were used to measure perceived price. The perceived benefit was measured using five items designed based on the study of [54]. Five items that were modified from [7] to assess purchase intention were used. The items in each variable are listed in Table 2. The recommended sample size for PLS-SEM is approximately 10 times the total number of paths leading towards variables [80]. The conceptual model has 9 arrows in the current study; however, the sample size is 429, which is larger than what is required. As a result, the condition of representativeness is met. As was already noted, the research sample is far larger than what is required in each country.

## 4. Validity of data and results

### 4.1. Validity of data and analysis

The descriptive analysis's results (Table 3) showed that survey participants from Kenya had the lowest perceptions of each construct in the conceptual model considered in this study. Next came the French respondents. Regarding the United States respondents, they had the highest scores of the three nations.

### 4.2. Common method bias (CMB), validity and reliability

A thorough study of collinearity was made to assess CMB using collinearity statistics (VIF). The test revealed that all collinearity inner values of latent variables are noticeably below the 3.3 cutoff point (Table 3). That indicates that the CMB is not an issue in this current research. The variables' reliability was assessed using Cronbach's Alpha and Composite Reliability (CR). The entire sample was first evaluated, and items with factor loadings of less than 0.600 were removed. In Table 4, the results of validity and reliability are shown together with the factor loadings of the remnant items for the sample of each nation and the entire sample. Alpha values and CR exceeded the advisable value of 0.700. The CRs supported convergent validity and Average Variance Extracted (AVE), which were all greater than or near 0.7 and 0.5, respectively. Cross-loading analysis was used to evaluate discriminant validity. The evaluation of the multicollinearity showed that the value of each indicator's Variance Inflation Factor (VIF) was

**Table 2. Scales items.**

| |
|---|
| ***Budget Perception- Forecasting (self-developed) [7, 26, 42, 70]*** |
| BG1: I plan the amount to spend for my smartphone (s) purchase on international online selling platforms. |
| BG2: I plan the financial amount I need for my smartphone (s) purchase on international online selling platforms as much as possible. |
| BG3: I always set in advance the approximate expenses I have to make for purchasing my smartphone (s) on international online selling platforms. |
| ***Budget Perception- Choice (self-developed) [7, 26, 42, 70]*** |
| BG4: My buying decision of smartphone (s) on international online selling platforms depends on my spending plan. |
| BG5: I evaluate smartphone offers on international online selling platforms according to my spending plan. |
| BG6: I compare different offers of smartphone offers on international online selling platforms according to my needs and financial plan. |
| BG7: My spending plan assists me in evaluating and motivating my purchase decision-making of smartphones on international online selling platforms. |
| BG8: My spending plan assists me in choosing among different proposals when buying smartphones on international online selling platforms. |
| ***Perceived quality (Wen and Fang 2014) [71]*** |
| PPQ1: I think smartphones sold on international online selling platforms are of superior quality. |
| PPQ2: The smartphone (s) that I bought on international online selling platforms is of equal quality to those in the physical stores. |
| PPQ3: The smartphone (s) that I bought on international online selling platforms was as I expected. |
| PPQ4: I am satisfied with the smartphone (s) I bought on international online selling platforms. |
| ***Perceived price (Kim et al. 2010) [49]*** |
| PPR1: Buying smartphone (s) on international online selling platforms may be more expensive than local marketplaces. |
| PPR2: I will probably save more money buying smartphone (s) on local marketplaces than on international online selling platforms. |
| PPR3: It may be possible to get a better discount when buying smartphone (s) on local marketplaces than on international online selling platforms. |
| PPR4: It may be cheaper to buy a smartphone (s) on international online selling platforms than at local marketplaces. |
| ***Perceived benefit (Yang et al. 2020) [54]*** |
| PB1: I get more information about the smartphone (s) when purchasing on international online selling platforms. |
| PB2: The smartphone price is much lower via international online selling platforms than local marketplaces. |
| PB3: International online selling platforms provide me with great flexibility in shopping. |
| PB4: I can get original smartphones with less effort, time, and money on international online selling platforms. |
| PB5: I achieved more satisfaction purchasing smartphone (s) through international online selling platforms. |
| ***Purchase Intention (Bringula et al. 2018) [7]*** |
| PIT1: If I have to buy a new smartphone, I will buy it from international online selling platforms. |
| PIT2: I am willing to use my credit or debit card to buy a smartphone from international online selling platforms. |
| PIT3: I am willing to give my personal information to buy a smartphone from international online selling platforms. |
| PIT4: I am willing to pay a delivery fee on any international online selling platform. |
| PIT5: I am willing to wait for the delivery of the smartphone (s) bought from international online selling platforms. |

less than 5. Additionally, the Heterotrait-Monotrait Method (HTMT) and the criteria proposed by Fornell and Larcker were utilized to assess the discriminant validity of the data. Table 5 lists the outcomes of both analyses. Table 6 summarizes the cross-factor loadings for each item. A characteristic of discriminant validity is the fact that each of the factor loadings was greater than their cross-loadings.

**Table 3. Descriptive analysis.**

| Country | Average Age | Survey received | Valid Survey | Usable Rate (%) | Men Rate (%) | Women Rate (%) |
|---|---|---|---|---|---|---|
| France | 30.114 | 191 | 149 | 78.010 | 52.348 | 47.652 |
| Kenya | 29.201 | 201 | 139 | 69.154 | 47.482 | 52.518 |
| United states | 28.894 | 178 | 141 | 79.213 | 46.099 | 53.901 |
| Total | 29.417 | 570 | 429 | 75.45666667 | 48.96666667 | 51.36333333 |

| | Kenya | | France | | United States | | Total sample | |
|---|---|---|---|---|---|---|---|---|
| | Mean | SD | Mean | SD | Mean | SD | Mean | SD |
| **BGT** | | | | | | | | |
| BGT1 | 3.597 | 0.903 | 3.483 | 0.994 | 3.305 | 1.16 | 3.462 | 1.032 |
| BGT2 | 3.835 | 0.894 | 3.732 | 0.96 | 3.56 | 1.163 | 3.709 | 1.018 |
| BGT3 | 3.381 | 1.048 | 3.403 | 1.111 | 3.014 | 1.155 | 3.268 | 1.12 |
| BGT4 | 3.662 | 0.941 | 3.638 | 0.964 | 3.404 | 1.161 | 3.569 | 1.032 |
| BGT5 | 3.396 | 0.942 | 3.436 | 1.032 | 3.184 | 1.152 | 3.34 | 1.051 |
| BGT6 | 2.986 | 0.952 | 3.107 | 1.075 | 2.702 | 1.103 | 2.935 | 1.06 |
| BGT7 | 3.273 | 0.896 | 3.396 | 0.989 | 2.823 | 1.169 | 3.168 | 1.053 |
| BGT8 | 3.676 | 0.891 | 3.738 | 1.006 | 3.376 | 1.152 | 3.599 | 1.034 |
| **PB** | | | | | | | | |
| PB1 | 2.813 | 0.735 | 2.819 | 0.803 | 2.879 | 1.062 | 2.837 | 0.878 |
| PB2 | 2.698 | 0.837 | 2.691 | 0.835 | 2.823 | 1.156 | 2.737 | 0.955 |
| PB3 | 3.050 | 0.771 | 3.141 | 0.934 | 3.298 | 1.037 | 3.163 | 0.927 |
| PB4 | 2.597 | 0.784 | 2.826 | 0.88 | 2.887 | 1.137 | 2.772 | 0.953 |
| PB5 | 2.331 | 0.781 | 2.510 | 0.938 | 2.716 | 1.228 | 2.52 | 1.012 |
| **PIT** | | | | | | | | |
| PIT1 | 3.007 | 0.818 | 3.040 | 0.776 | 2.816 | 0.919 | 2.956 | 0.845 |
| PIT2 | 3.101 | 0.884 | 3.087 | 0.897 | 2.922 | 1.004 | 3.037 | 0.933 |
| PIT3 | 3.115 | 0.937 | 3.181 | 0.867 | 2.929 | 1.015 | 3.077 | 0.947 |
| PIT4 | 3.173 | 0.929 | 3.081 | 0.871 | 2.865 | 1.06 | 3.04 | 0.964 |
| PIT5 | 2.986 | 0.805 | 2.933 | 0.988 | 2.773 | 1.027 | 2.897 | 0.95 |
| **PPQ** | | | | | | | | |
| PPQ1 | 2.108 | 0.546 | 2.148 | 0.736 | 2.348 | 0.982 | 2.2 | 0.783 |
| PPQ2 | 2.338 | 0.641 | 2.497 | 0.738 | 2.766 | 0.95 | 2.534 | 0.806 |
| PPQ3 | 2.482 | 0.682 | 2.383 | 0.641 | 2.532 | 0.896 | 2.464 | 0.749 |
| PPQ4 | 2.345 | 0.643 | 2.403 | 0.741 | 2.631 | 0.971 | 2.459 | 0.806 |
| **PPR** | | | | | | | | |
| PPR1 | 1.899 | 0.884 | 1.913 | 0.897 | 2.078 | 1.004 | 1.963 | 0.933 |
| PPR2 | 1.885 | 0.937 | 1.819 | 0.867 | 2.071 | 1.015 | 1.923 | 0.947 |
| PPR3 | 1.827 | 0.929 | 1.919 | 0.871 | 2.135 | 1.06 | 1.96 | 0.964 |
| PPR4 | 2.014 | 0.805 | 2.067 | 0.988 | 2.227 | 1.027 | 2.103 | 0.95 |

| Collinearity Statistics (VIF) | | | | | | | |
|---|---|---|---|---|---|---|---|
| | BGT | PB | PIT | PPQ | PPR | | |
| BGT | 0.000 | 1.204 | 1.446 | 1.204 | 1 | | |
| PB | 0.000 | 0.000 | 1.679 | 0.000 | 0.000 | | |
| PIT | 0.000 | 0.000 | 0.000 | 0.000 | 0.000 | | |
| PPQ | 0.000 | 0.000 | 1.984 | 0.000 | 0.000 | | |
| PPR | 0.000 | 1.204 | 1.672 | 1.204 | 0.000 | | |

**Table 4. Factor loadings, reliability and validity.**

| | Kenya | | | | | France | | | | | United States | | | | | Total sample | | | | |
|---|---|---|---|---|---|---|---|---|---|---|---|---|---|---|---|---|---|---|---|---|
| | FL | Alpha | GR | AVE | VIF | FL | Alpha | GR | AVE | VIF | FL | Alpha | GR | AVE | VIF | FL | Alpha | GR | AVE | VIF |
| BGT1 | 0.574 | 0.841 | 0.878 | 0.475 | 1.501 | 0.72 | 0.871 | 0.899 | 0.527 | 1.81 | 0.679 | 0.879 | 0.904 | 0.542 | 1.682 | 0.672 | 0.871 | 0.899 | 0.527 | 1.598 |
| BGT2 | 0.629 | | | | 1.545 | 0.654 | | | | 1.488 | 0.699 | | | | 1.867 | 0.671 | | | | 1.595 |
| BGT3 | 0.762 | | | | 2.256 | 0.779 | | | | 2.305 | 0.751 | | | | 2.047 | 0.769 | | | | 2.159 |
| BGT4 | 0.632 | | | | 1.389 | 0.629 | | | | 1.455 | 0.655 | | | | 1.523 | 0.643 | | | | 1.427 |
| BGT5 | 0.738 | | | | 1.92 | 0.766 | | | | 2.235 | 0.787 | | | | 2.311 | 0.768 | | | | 2.06 |
| BGT6 | 0.737 | | | | 2.239 | 0.773 | | | | 2.264 | 0.72 | | | | 2.071 | 0.748 | | | | 2.099 |
| BGT7 | 0.782 | | | | 2.675 | 0.793 | | | | 2.575 | 0.761 | | | | 2.375 | 0.786 | | | | 2.484 |
| BGT8 | 0.632 | | | | 1.854 | 0.674 | | | | 1.796 | 0.825 | | | | 2.671 | 0.739 | | | | 2.025 |
| PB1 | 0.547 | 0.749 | 0.773 | 0.409 | 1.16 | 0.596 | 0.774 | 0.843 | 0.52 | 1.344 | 0.692 | 0.853 | 0.895 | 0.632 | 1.593 | 0.636 | 0.798 | 0.86 | 0.553 | 1.361 |
| PB2 | 0.738 | | | | 1.254 | 0.774 | | | | 1.405 | 0.823 | | | | 2.18 | 0.776 | | | | 1.564 |
| PB3 | 0.596 | | | | 1.278 | 0.825 | | | | 1.880 | 0.817 | | | | 1.966 | 0.779 | | | | 1.711 |
| PB4 | 0.543 | | | | 1.238 | 0.661 | | | | 1.468 | 0.769 | | | | 1.719 | 0.701 | | | | 1.505 |
| PB5 | 0.743 | | | | 1.375 | 0.728 | | | | 1.567 | 0.863 | | | | 2.411 | 0.812 | | | | 1.775 |
| PIT1 | 0.371 | 0.715 | 0.813 | 0.479 | 1.176 | 0.573 | 0.717 | 0.813 | 0.467 | 1.239 | 0.759 | 0.815 | 0.871 | 0.575 | 1.72 | 0.61 | 0.765 | 0.841 | 0.515 | 1.316 |
| PIT2 | 0.764 | | | | 1.562 | 0.692 | | | | 1.361 | 0.778 | | | | 1.761 | 0.75 | | | | 1.548 |
| PIT3 | 0.784 | | | | 1.795 | 0.696 | | | | 1.393 | 0.755 | | | | 1.771 | 0.75 | | | | 1.63 |
| PIT4 | 0.796 | | | | 1.636 | 0.719 | | | | 1.387 | 0.774 | | | | 1.645 | 0.766 | | | | 1.538 |
| PIT5 | 0.652 | | | | 1.378 | 0.726 | | | | 1.406 | 0.724 | | | | 1.541 | 0.7 | | | | 1.406 |
| PPQ1 | 0.758 | 0.744 | 0.789 | 0.486 | 1.341 | 0.807 | 0.761 | 0.847 | 0.582 | 1.489 | 0.798 | 0.857 | 0.903 | 0.7 | 1.756 | 0.796 | 0.797 | 0.868 | 0.623 | 1.584 |
| PPQ2 | 0.607 | | | | 1.317 | 0.79 | | | | 1.652 | 0.84 | | | | 2.12 | 0.791 | | | | 1.752 |
| PPQ3 | 0.645 | | | | 1.208 | 0.662 | | | | 1.315 | 0.811 | | | | 1.827 | 0.721 | | | | 1.422 |
| PPQ4 | 0.764 | | | | 1.471 | 0.785 | | | | 1.667 | 0.896 | | | | 2.722 | 0.844 | | | | 1.996 |
| PPR1 | 0.761 | 0.747 | 0.841 | 0.57 | 1.562 | 0.683 | 0.688 | 0.809 | 0.515 | 1.332 | 0.788 | 0.77 | 0.853 | 0.592 | 1.75 | 0.751 | 0.744 | 0.838 | 0.565 | 1.538 |
| PPR2 | 0.762 | | | | 1.636 | 0.675 | | | | 1.323 | 0.736 | | | | 1.587 | 0.729 | | | | 1.501 |
| PPR3 | 0.809 | | | | 1.629 | 0.758 | | | | 1.37 | 0.806 | | | | 1.602 | 0.795 | | | | 1.519 |
| PPR4 | 0.682 | | | | 1.364 | 0.751 | | | | 1.395 | 0.744 | | | | 1.416 | 0.73 | | | | 1.391 |

## 4.3. Structural model

We then evaluated the proposed hypothesized links in our conceptual model. Direct relationships were assessed first. Table 7 presents the outcome of that assessment. Table 7 presents the results for the entire sample as well as the findings for each nation. The results shown in Table 7 demonstrate that all other hypotheses were either negatively significant or insignificant, with the exception of H4: PPR-> PPQ (b = 0.408, t = 10.109, p = 0.000) and H5: PPR-> PB (b = 0.476, t = 10.273, p = 0.000). As a result, whereas H1, H2a, H2b, H2c, H3a, H3b, and H3c are rejected, only hypotheses H5 and H6 are accepted. Also, results show that samples from each country are largely consistent with those for the entire sample. For instance, in the Kenya, France and United states samples, the findings show that most hypotheses were likewise negatively significant, with the exception of H4:PPR -> PPQ (Kenya(b = 0.385, t = 4.328, p = 0.000); France ((b = 0.413, t = 5.951, p = 0.000)); United States ((b = 0.431, t = 7.276, p = 0.000)) and H5:PPR -> PB (Kenya(b = 0.479, t = 7.017, p = 0.000); France ((b = 0.376, t = 4.505, p = 0.000)); United States ((b = 0.569, t = 8.003, p = 0.000)). Therefore, with each country's sample, hypotheses H1, H2a, H2b, H2c, H3a, H3b, and H3c are rejected, while hypotheses H5 and H6 are accepted.

**Table 5. Discriminant validity using Heterotrait-Monotrait (HTMT) and Fornell & Larcker (LK).**

|  | BGT | PB | PIT | PPQ | PPR |
|---|---|---|---|---|---|
| Kenya |  |  |  |  |  |
| BGT | *0.69* | 0.462 | 0.432 | 0.508 | 0.417 |
| PB | -0.353 | *0.64* | 0.727 | 0.616 | 0.718 |
| PIT | 0.356 | -0.542 | *0.692* | 0.741 | 1.313 |
| PPQ | -0.389 | 0.408 | -0.489 | *0.697* | 0.681 |
| PPR | -0.353 | 0.544 | -0.994 | 0.474 | *0.755* |
| France |  |  |  |  |  |
| BGT | *0.726* | 0.437 | 0.599 | 0.675 | 0.561 |
| PB | -0.383 | *0.721* | 0.568 | 0.631 | 0.579 |
| PIT | 0.479 | -0.469 | *0.683* | 0.803 | 1.373 |
| PPQ | -0.568 | 0.509 | -0.604 | *0.763* | 0.78 |
| PPR | -0.456 | 0.472 | -0.984 | 0.586 | *0.718* |
| United States |  |  |  |  |  |
| BGT | *0.736* | 0.393 | 0.479 | 0.65 | 0.463 |
| PB | -0.356 | *0.795* | 0.756 | 0.752 | 0.75 |
| PIT | 0.415 | -0.636 | *0.758* | 0.718 | 1.236 |
| PPQ | -0.571 | 0.653 | -0.601 | *0.837* | 0.716 |
| PPR | -0.399 | 0.62 | -0.984 | 0.59 | *0.769* |
| Overall sample |  |  |  |  |  |
| BGT | *0.726* | 0.423 | 0.519 | 0.639 | 0.495 |
| PB | -0.369 | *0.744* | 0.691 | 0.707 | 0.686 |
| PIT | 0.427 | -0.554 | *0.718* | 0.746 | 1.285 |
| PPQ | -0.538 | 0.574 | -0.579 | *0.789* | 0.721 |
| PPR | -0.412 | 0.547 | -0.985 | 0.561 | *0.752* |

## 4.4. The analysis of the mediators

The results of the mediation links for each country and the total sample are shown in Table 7 (below). PPR (b = 0.395, t = 8.789, p>0.001) and PPQ (b = 0.011, t = 2.478, p>0.05) showed significantly positive mediating roles, according to the results for the entire sample, although PB (= 0.001, t = 0.584, p = 0.559) did not. As a result, hypotheses H4a and H4b are confirmed, and hypothesis H4c is refuted. Additionally, PB showed an insignificant mediating role for the samples of Kenya ((b = -0.001, t = 0.239, p = 0.811); France (b = -0.002, t = 0.368, p = 0.713) and the United States (b = 0.005, t = 1.029, p = 0.303). Therefore, hypothesis H4c failed for each group-specific (country). Concerning the mediating role of PPR, each nation results are in line with the results for the entire sample. For example, with the samples of Kenya, France and the United States, PPR (Kenya (= 0.347, t = 4.393, p> 0.001)), France (b = 0.437, t = 6.976, p>0.001) and the United States (= 0.437, t = 6.976, p>0.001)) showed significant and positive mediating roles. However, the mediating role of PPQ was insignificant according to the samples of Kenya (b = 0.006, t = 1.394, p = 0.163), France (b = 0.012, t = 1.208, p = 0.227), and the United States (b = 0.002, t = 0.176, p = 0.86), whereas it was significantly positive according to the overall sample, as already stated above. Therefore, hypotheses H4a and H4c are rejected, while hypothesis H4b is accepted in each country-specific sample.

## 4.5. Multi-group analysis

In the study's last phase, we evaluated the variations in BGT's effects in the nations covered in this research, i.e., Kenya, France, and the United States. The multi-group analysis was used to

**Table 6. Discriminant validity- cross loading.**

| | Kenya | | | | | France | | | | | United states | | | | | Total sample | | | | |
|---|---|---|---|---|---|---|---|---|---|---|---|---|---|---|---|---|---|---|---|---|
| | BGT | PB | PIT | PPQ | PPR | BGT | PB | PIT | PPQ | PPR | BGT | PB | PIT | PPQ | PPR | BGT | PB | PIT | PPQ | PPR |
| BGT1 | 0.574 | -0.15 | 0.182 | -0.192 | -0.183 | 0.72 | -0.405 | 0.339 | -0.445 | -0.338 | 0.679 | -0.221 | 0.237 | -0.391 | -0.234 | 0.672 | -0.271 | 0.263 | -0.374 | -0.262 |
| BGT2 | 0.629 | -0.291 | 0.257 | -0.266 | -0.264 | 0.654 | -0.308 | 0.429 | -0.501 | -0.403 | 0.699 | -0.338 | 0.379 | -0.482 | -0.384 | 0.671 | -0.328 | 0.37 | -0.446 | -0.366 |
| BGT3 | 0.762 | -0.173 | 0.279 | -0.263 | -0.265 | 0.779 | -0.193 | 0.271 | -0.483 | -0.241 | 0.751 | -0.219 | 0.252 | -0.404 | -0.245 | 0.769 | -0.206 | 0.281 | -0.403 | -0.262 |
| BGT4 | 0.632 | -0.334 | 0.351 | -0.32 | -0.346 | 0.629 | -0.314 | 0.411 | -0.331 | -0.399 | 0.655 | -0.311 | 0.321 | -0.395 | -0.298 | 0.643 | -0.32 | 0.365 | -0.364 | -0.351 |
| BGT5 | 0.738 | -0.206 | 0.233 | -0.295 | -0.24 | 0.766 | -0.19 | 0.324 | -0.366 | -0.315 | 0.787 | -0.241 | 0.271 | -0.444 | -0.257 | 0.768 | -0.217 | 0.284 | -0.392 | -0.279 |
| BGT6 | 0.737 | -0.219 | 0.265 | -0.317 | -0.255 | 0.773 | -0.253 | 0.32 | -0.392 | -0.302 | 0.72 | -0.238 | 0.214 | -0.342 | -0.214 | 0.748 | -0.24 | 0.278 | -0.361 | -0.267 |
| BGT7 | 0.782 | -0.268 | 0.165 | -0.174 | -0.166 | 0.793 | -0.282 | 0.349 | -0.391 | -0.327 | 0.761 | -0.27 | 0.365 | -0.439 | -0.345 | 0.786 | -0.284 | 0.326 | -0.389 | -0.31 |
| BGT8 | 0.632 | -0.241 | 0.104 | -0.235 | -0.099 | 0.674 | -0.202 | 0.269 | -0.328 | -0.257 | 0.825 | -0.222 | 0.341 | -0.421 | -0.309 | 0.739 | -0.23 | 0.27 | -0.365 | -0.25 |
| PB1 | -0.056 | 0.547 | -0.304 | 0.247 | 0.299 | -0.177 | 0.596 | -0.182 | 0.197 | 0.172 | -0.153 | 0.692 | -0.383 | 0.394 | 0.372 | -0.138 | 0.636 | -0.306 | 0.308 | 0.296 |
| PB2 | -0.286 | 0.738 | -0.473 | 0.233 | 0.469 | -0.323 | 0.774 | -0.489 | 0.452 | 0.489 | -0.253 | 0.823 | -0.524 | 0.605 | 0.507 | -0.284 | 0.776 | -0.503 | 0.484 | 0.494 |
| PB3 | -0.199 | 0.596 | -0.169 | 0.192 | 0.188 | -0.345 | 0.825 | -0.335 | 0.414 | 0.344 | -0.328 | 0.817 | -0.498 | 0.521 | 0.494 | -0.316 | 0.779 | -0.367 | 0.426 | 0.373 |
| PB4 | -0.247 | 0.543 | -0.231 | 0.206 | 0.237 | -0.279 | 0.661 | -0.223 | 0.339 | 0.217 | -0.239 | 0.769 | -0.519 | 0.443 | 0.517 | -0.261 | 0.701 | -0.364 | 0.371 | 0.361 |
| PB5 | -0.297 | 0.743 | -0.424 | 0.393 | 0.425 | -0.219 | 0.728 | -0.337 | 0.35 | 0.348 | -0.404 | 0.863 | -0.578 | 0.605 | 0.551 | -0.335 | 0.812 | -0.474 | 0.502 | 0.465 |
| PIT1 | 0.158 | -0.179 | 0.371 | -0.283 | -0.265 | 0.377 | -0.244 | 0.573 | -0.415 | -0.421 | 0.374 | -0.528 | 0.759 | -0.482 | -0.632 | 0.326 | -0.356 | 0.61 | -0.421 | -0.467 |
| PIT2 | 0.254 | -0.328 | 0.764 | -0.366 | -0.761 | 0.266 | -0.328 | 0.692 | -0.29 | -0.683 | 0.263 | -0.393 | 0.778 | -0.422 | -0.788 | 0.267 | -0.343 | 0.75 | -0.367 | -0.751 |
| PIT3 | 0.194 | -0.368 | 0.784 | -0.345 | -0.762 | 0.248 | -0.187 | 0.696 | -0.369 | -0.675 | 0.19 | -0.402 | 0.755 | -0.334 | -0.736 | 0.22 | -0.321 | 0.75 | -0.348 | -0.729 |
| PIT4 | 0.299 | -0.51 | 0.796 | -0.392 | -0.809 | 0.465 | -0.42 | 0.719 | -0.546 | -0.758 | 0.374 | -0.566 | 0.774 | -0.538 | -0.806 | 0.391 | -0.507 | 0.766 | -0.506 | -0.795 |
| PIT5 | 0.319 | -0.428 | 0.652 | -0.327 | -0.682 | 0.301 | -0.393 | 0.726 | -0.448 | -0.751 | 0.376 | -0.526 | 0.724 | -0.499 | -0.744 | 0.342 | -0.453 | 0.7 | -0.448 | -0.73 |
| PPQ1 | -0.368 | 0.231 | -0.349 | 0.758 | 0.342 | -0.571 | 0.403 | -0.54 | 0.807 | 0.517 | -0.511 | 0.417 | -0.447 | 0.798 | 0.429 | -0.509 | 0.389 | -0.459 | 0.796 | 0.441 |
| PPQ2 | -0.15 | 0.209 | -0.298 | 0.607 | 0.271 | -0.36 | 0.357 | -0.489 | 0.79 | 0.491 | -0.44 | 0.537 | -0.518 | 0.84 | 0.509 | -0.369 | 0.427 | -0.472 | 0.791 | 0.454 |
| PPQ3 | -0.325 | 0.292 | -0.324 | 0.645 | 0.325 | -0.405 | 0.308 | -0.346 | 0.662 | 0.326 | -0.417 | 0.566 | -0.524 | 0.811 | 0.509 | -0.39 | 0.438 | -0.42 | 0.721 | 0.407 |
| PPQ4 | -0.217 | 0.391 | -0.386 | 0.764 | 0.374 | -0.37 | 0.484 | -0.439 | 0.785 | 0.426 | -0.543 | 0.652 | -0.519 | 0.896 | 0.525 | -0.426 | 0.558 | -0.475 | 0.844 | 0.466 |
| PPR1 | -0.254 | 0.328 | -0.764 | 0.366 | 0.761 | -0.266 | 0.328 | -0.692 | 0.29 | 0.683 | -0.263 | 0.393 | -0.778 | 0.422 | 0.788 | -0.267 | 0.343 | -0.75 | 0.367 | 0.751 |
| PPR2 | -0.194 | 0.368 | -0.784 | 0.345 | 0.762 | -0.248 | 0.187 | -0.696 | 0.369 | 0.675 | -0.19 | 0.402 | -0.755 | 0.334 | 0.736 | -0.22 | 0.321 | -0.75 | 0.348 | 0.729 |
| PPR3 | -0.299 | 0.51 | -0.796 | 0.392 | 0.809 | -0.465 | 0.42 | -0.719 | 0.546 | 0.758 | -0.374 | 0.566 | -0.774 | 0.538 | 0.806 | -0.391 | 0.507 | -0.766 | 0.506 | 0.795 |
| PPR4 | -0.319 | 0.428 | -0.652 | 0.327 | 0.682 | -0.301 | 0.393 | -0.726 | 0.448 | 0.751 | -0.376 | 0.526 | -0.724 | 0.499 | 0.744 | -0.342 | 0.453 | -0.7 | 0.448 | 0.73 |

**Table 7. Direct links between variables and mediation analysis.**

| Direct effect | Kenya | | | France | | | United states | | | Total sample | | |
|---|---|---|---|---|---|---|---|---|---|---|---|---|
| | B | T | P | B | T | P | B | T | P | B | T | P |
| H1: BGT -> PIT | 0.001 | 0.098 | 0.922 | 0.028 | 1.263 | 0.207 | 0.020 | 1.089 | 0.276 | 0.014 | 1.269 | 0.205 |
| H2a: BGT -> PPQ | -0.253 | 2.924 | 0.003 | -0.379 | 5.609 | 0.000 | -0.399 | 5.934 | 0.000 | -0.370 | 8.558 | 0.000 |
| H2b: BGT -> PPR | -0.353 | 4.409 | 0.000 | -0.456 | 6.937 | 0.000 | -0.399 | 4.636 | 0.000 | -0.412 | 8.721 | 0.000 |
| H2c: BGT -> PB | -0.184 | 2.516 | 0.012 | -0.211 | 2.266 | 0.024 | -0.130 | 1.472 | 0.141 | -0.172 | 3.119 | 0.002 |
| H3a: PPQ -> PIT | -0.023 | 1.838 | 0.066 | -0.032 | 1.265 | 0.206 | -0.004 | 0.183 | 0.855 | -0.030 | 2.620 | 0.009 |
| H3b: PPR -> PIT | -0.984 | 80.448 | 0.000 | -0.957 | 56.585 | 0.000 | -0.950 | 68.872 | 0.000 | -0.959 | 121.126 | 0.000 |
| H3c: PB -> PIT | 0.003 | 0.271 | 0.786 | 0.010 | 0.434 | 0.665 | -0.037 | 1.730 | 0.084 | -0.007 | 0.618 | 0.536 |
| H5: PPR -> PPQ | 0.385 | 4.328 | 0.000 | 0.413 | 5.951 | 0.000 | 0.431 | 7.276 | 0.000 | 0.408 | 10.109 | 0.000 |
| H6: PPR -> PB | 0.479 | 7.017 | 0.000 | 0.376 | 4.505 | 0.000 | 0.569 | 8.003 | 0.000 | 0.476 | 10.273 | 0.000 |
| | Kenya | | | France | | | United states | | | Overall sample | | |
| | B | T | P | B | T | P | B | T | P | B | T | P |
| Specific Indirect effect | | | | | | | | | | | | |
| H4a: BGT -> PPQ -> PIT | 0.006 | 1.394 | 0.163 | 0.012 | 1.208 | 0.227 | 0.002 | 0.176 | 0.86 | 0.011 | 2.478 | 0.013 |
| H4b: BGT -> PPR -> PIT | 0.347 | 4.393 | 0.000 | 0.437 | 6.976 | 0.000 | 0.379 | 4.625 | 0.000 | 0.395 | 8.789 | 0.000 |
| H4c: BGT -> PB -> PIT | -0.001 | 0.239 | 0.811 | -0.002 | 0.368 | 0.713 | 0.005 | 1.029 | 0.303 | 0.001 | 0.584 | 0.559 |
| Total effect | | | | | | | | | | | | |
| BGT -> PIT | 0.356 | 4.475 | 0.000 | 0.479 | 7.338 | 0.000 | 0.415 | 4.813 | 0.000 | 0.428 | 8.995 | 0.000 |

evaluate the H7 hypothesis (Table 8). The result showed that the variations were not especially apparent when contrasting BGT's impact between countries. Therefore, our data did not support hypothesis H8a. These outcomes demonstrate that customer perception of the purchase budgets across Kenya, France, and the United States are quite similar.

## 5. Discussion

The study explored the direct and mediated impacts of customer perception of the purchase budget (BGT), product perceived quality (PPQ), perceived price (PPR), and perceived benefit (PB) in a cross-country environment (Kenya, France, and the United States). A positive mediating role of PPQ between BGT and PIT was found for the entire sample but did not mediate the samples country-specific. That result is in line with the planned behaviour theory, which backs up the claim that customers may identify with products that are related to their purchase

**Table 8. Multi-group comparison (MGA) (H7).**

| Path Difference | Diff. (Kenya—United states) | p-Value (Kenya vs United states) | Diff. (France—Kenya) | p-Value (France vs Kenya) | Diff. (France—United states) | p-Value (France vs United states) |
|---|---|---|---|---|---|---|
| BGT -> PB | -0.054 | 0.686 | -0.027 | 0.589 | -0.081 | 0.738 |
| BGT -> PIT | -0.019 | 0.800 | 0.026 | 0.149 | 0.007 | 0.398 |
| BGT -> PPQ | 0.145 | 0.088 | -0.126 | 0.878 | 0.019 | 0.419 |
| BGT -> PPR | 0.046 | 0.342 | -0.104 | 0.843 | -0.058 | 0.700 |

and expend plan. PPR was found to significantly mediate between BGT and PIT in both group-specific samples (Kenya, France, and the United States) and the overall sample. This outcome shows that perception of the purchase budget seems highly important during the formation of the customer purchase intention and in evaluating smartphone quality and price options. The findings may be understood based on the argument that perceived price is the key factor guiding consumer spending plans and purchase intention. Online sellers and smartphone brands now need to understand that the perception customers may have of the price has a link with their perception of the spending plans and purchase intention. This understanding might greatly assist them in enhancing their offer in terms of price to boost sales. The findings revealed PPR's positive and significant effect on PPQ and PB. These impacts may be linked to PPR's role in sending favorable indications to the consumers regarding the expected product quality, which would assist in developing a positive perceived benefit [86]. These positive correlations have previously been demonstrated by earlier investigations [63, 68, 70]. As such, our results are consistent with earlier research. Research from the past supports the importance of PPR on PPQ and PB. Previous studies strongly support the idea that consumers' perceptions of price and product quality are strongly correlated [70]. That is also a crucial component of consumers' perceived benefit when shopping online [63]. Similar to this, research pointed out that customers purchase more expensive products if the perceived benefit justifies the expenditure [51]. Contrarily, the results of prior studies indicated that PPQ and PPR had a positive and significant influence on PIT [19, 55]. This study shows that relationships between PPQ, PPR, and PIT were either negative or insignificant. For instance, the findings showed that PPQ significantly and negatively affects PIT in the overall sample, while the effect was insignificant in group-specific samples (Kenya, France, and the United States). The insignificant effect of PPQ on PIT may be explained by the fact that purchasing a smartphone online differs from the purchase intention since smartphone quality attributes are not assessed on shopping platforms but rather outside [7]. Additionally, the study also showed that the relationship between PPR and PIT was negatively significant in the group-specific samples and overall. The negative and significant effect of PPR on PIT is also corroborated by earlier work [19]. Indeed, customers may find it more challenging to make a purchase when they believe the product's pricing is excessively high [19]. Customers' perceived prices, therefore, negatively influence their purchase intention [17]. However, contrary to our result, [19] discovered a positive connection between PPR and PIT. Therefore, further investigation is required to determine how the links between PPR and PIT are built. As already stated, among the three mediators of the study, only PPR mediated the relationship between BGT and PIT in both country-specific samples and overall, while PPQ mediated only the whole sample. However, PB did not mediate the BGT-PIT link for both the whole and country-specific samples. Our findings show that for online sellers of smartphone brands to improve sales on international online shopping platforms, they should be more focused on improving pricing and using strategies that support product pricing in correlation with the perception of customer budgets. Doing so will create a positive perceived price in buyers' minds, which will increase the purchase's likelihood [61]. That is true even though previous research on smartphone purchase intention in e-shopping has largely ignored the role of PPR as a potential mediator between BGT and PIT. The results of evaluating the conceptual model in each nation's unique environment were remarkably consistent in most cases with the sample as a whole. Significant similarities, for instance, were found between the three nations. i.e., between the relationships of BGT and PPQ, and PPR; PPR and PIT; PPR and PPQ; PPR and PB. Significantly positive effects of PPR on PPQ and PB were found in each of the samples. In all three samples, it was discovered that the impact of PPR as a mediating factor in the BGT-PIT link replicated the overall result. That demonstrates some degree of uniformity in how customers perceive the purchase budget (BGT).

## 5.1. Managerial and practical implications

This study's results will inform smartphone brand managers and CBEC e-sellers of the significance of links between customer perceptions of their purchase budget and purchase intention. Therefore, this research has substantial practical implications for e-sellers and smartphone brands that sell their products on international online platforms. First, it will guide them in implementing their sales strategies to attract more customer attention by offering products in line with the perceptions of the values customers wish to find in the product and their perceptions of the purchase budget. That would necessarily improve the customers' purchase intentions. Second, e-sellers and smartphone brands thus must focus on smartphone offers that consider the standard of living. That would strengthen the links between the perception of the purchase budget and the perceptions of the critical values customers i. e., price, quality and benefits customers wish to find in the product. Doing so will strengthen customer purchase intention and improve the product's positive image in the customer's mind [87]. As [26] stated, buyers are generally more attracted to products when they perceive that the product could match their purchasing power. Their purchasing power supports their perception of the purchase budget. The perceived purchase budget appears as an essential factor which may be positively associated with the perceptions of the values that the customer is looking for in the product [27, 29] and positively associated with the customer's purchase intention, as demonstrated in this current study. Smartphone industries should focus on product offers that strengthen the links between the perception of values that customers wish to find in the product they purchase and the perception of purchase budget. That would allow e-sellers to offer customers products matching customer purchase budgets.

This research has identified that PPR and PPQ significantly mediate between BGT and PIT in smartphones purchased on CBEC platforms. Consequently, the smartphone industry must give more importance to PPR and PPQ in their product design strategies by offering products that match customer perception in terms of price, quality and purchase budget to succeed easily in that market. Scholars have already pointed out that it is essential to implement that kind of strategy to ensure success within the CBEC marketplace [7, 78]. For instance, [88] concluded that smartphone brands should focus on more sophisticated quality attributes in the smartphone design they propose to the market since, because of social media, customer requirements in terms of quality will increase in future. In this direction, [27] showed that more than 90% of new products are unsuccessful in marketplaces due to quality and pricing attributes that do not match customer expectations and perceptions. Therefore, managers and smartphone manufacturers must establish a bridge between the pricing and quality expectations as well as perception, and the perception of the purchase budget, which will lead to a positive purchase intention.

## 5.2. Implications for theory

The study's conclusions based on the total effect demonstrate that consumers consider the idea of the BGT since it was discovered to have a substantial impact of the BGT on PIT (without the presence of any mediator) in each country-specific sample and overall. It is also the case when considering the mediator PPR in each country-specific sample and overall. This suggests that consumer perception homogeneity toward BGT is responsible for aligning customer attitudes across different geographic locations. However, a noteworthy finding that varies among entire sample results concerns the insignificant mediating role of PPQ in the link between BGT and PIT in the samples from Kenya, France, and the United States but is significant in the overall sample. PPQ mediator's overall impact on the relationship of BGT and PIT is the sum of the minor influences of PPQ through each subsample. There is a variation

between PPQ mediator impact in the relationship of BGT and PIT from the overall sample to each country's sample. When all of the responses are combined, the impact of the PPQ mediator becomes significant due to this aggregation effect. In this approach, smartphone e-sellers on international online platforms may manage customers more wisely. That will improve the likelihood of customers' purchases. Smartphone e-sellers may use customer purchase budgets as a strategic tool to manage their connections with customers since they can easily propose customers' products in line with their budget. This research established a perspective to understand the smartphone's buying budget affects buyer purchase intention in smartphone selling through international online platforms. This approach will also help smartphone sellers, and brands better understand consumer behaviour and how they perceive purchase budgets (BGT) to develop effective marketing strategies.

### 5.3. Study's limitations, future research and conclusion

This study has some limitations. First, only three nations were examined in this study (Kenya, France, and the United States). Future research should include other nations with different economic backgrounds to further study the influence of BGT on customer purchase intention. Second, cross-sectional survey data was used to assess the hypothesis that underpinned the study. Future research may concentrate on longitudinal data showing how BGT, PPQ, PPR, PB, and PIT are changing dynamically. Third, future research might determine the mediating role of other factors, such as e-seller trust or credibility, in the relationship between BGT and PIT. Additionally, despite the fact that PPR impacted PB, our conceptual model revealed that PB was an insignificant mediator. Therefore, PB should be included as a possible mediator in future research to continue researching its involvement in comparable conceptual models and attempt to come to definitive conclusions regarding the mediating role of PB in the relationship between BGT and PIT. Fourthly, no moderating factors' impact was examined in this study. Future research might evaluate the moderating impact of smartphone brand popularity in each nation or brand availability in the local market.

### Supporting information

**S1 Dataset.**
(RAR)

### Acknowledgments

The authors acknowledge the editors and the anonymous reviewers for their insightful comments.

### Author Contributions

**Conceptualization:** Karamoko N'da.

**Data curation:** Karamoko N'da.

**Formal analysis:** Karamoko N'da.

**Funding acquisition:** Jiaoju Ge.

**Investigation:** Karamoko N'da.

**Methodology:** Karamoko N'da.

**Software:** Karamoko N'da.

**Supervision:** Jiaoju Ge, Steven Ji-Fan Ren, Jia Wang.

**Validation:** Jiaoju Ge, Steven Ji-Fan Ren.

**Visualization:** Karamoko N'da.

**Writing – original draft:** Karamoko N'da.

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
