## [Decision Letter · Decision Letter 0]

14 Nov 2022

PONE-D-22-28268Purchase budget (BGT) and purchase intention in smartphone selling industry: A cross-country analysisPLOS ONE

Dear Dr. N'da,

Thank you for submitting your manuscript to PLOS ONE. After careful consideration, we feel that it has merit but does not fully meet PLOS ONE’s publication criteria as it currently stands. Therefore, we invite you to submit a revised version of the manuscript that addresses the points raised during the review process.

We look forward to receiving your revised manuscript.

Kind regards,

Mingyue Fan, Ph.D.

Academic Editor

PLOS ONE

2. Thank you for including your ethics statement:  "N/A".  

a. For studies reporting research involving human participants, PLOS ONE requires authors to confirm that this specific study was reviewed and approved by an institutional review board (ethics committee) before the study began. Please provide the specific name of the ethics committee/IRB that approved your study, or explain why you did not seek approval in this case.

b. Please provide additional details regarding participant consent. In the ethics statement in the Methods and online submission information, please ensure that you have specified (1) whether consent was informed and (2) what type you obtained (for instance, written or verbal, and if verbal, how it was documented and witnessed). If your study included minors, state whether you obtained consent from parents or guardians. If the need for consent was waived by the ethics committee, please include this information.

4. Please include your tables as part of your main manuscript and remove the individual files. Please note that supplementary tables (should remain/ be uploaded) as separate "supporting information" files

Reviewers' comments:

Reviewer's Responses to Questions

**Comments to the Author**

1. Is the manuscript technically sound, and do the data support the conclusions?

Reviewer #1: Yes

Reviewer #2: Yes

2. Has the statistical analysis been performed appropriately and rigorously? 

Reviewer #1: Yes

Reviewer #2: Yes

3. Have the authors made all data underlying the findings in their manuscript fully available?

Reviewer #1: Yes

Reviewer #2: No

4. Is the manuscript presented in an intelligible fashion and written in standard English?

Reviewer #1: Yes

Reviewer #2: Yes

5. Review Comments to the Author

Reviewer #1: Yes empirically I agree the claim on mediating role between BGT and PIT but how do you fit in the theory of planned behavior needs more elaboration.

Why only Kenya, France and US? Why not compare other emerging countries with US and France? Is Kenya leading the way in cell phone usage? Please compare its facts and figures with other regional countries.

The results are pretty similar for the three countries. Don’t you think the purchasing power and culture may cause result differentials?

There is confusion whether purchase budget or perceived purchase budget is under consideration. Title and hypothesis tell different story.

Please read and cite the following papers:

10.7176/EJBM/11-12-18

DOI 10.3389/fenvs.2022.973692

Reviewer #2: Thank you so much for providing me with the opportunity to review the manuscript. I appreciate all authors who put in the effort and came up with a good piece of research. However, I am looking at significant matters requiring justifications to improve the quality of the paper overall.

1. Introduction.

- Can you elaborate on the specific problem that triggers you to conduct this study? Though you have mentioned some literature gaps which are very fine but I am looking towards the specific practical issue esp in three chosen countries.

- In line 108: You mentioned "The research will advance knowledge 109 of BGT-PIT links in cross-national contexts" ...

how it advances and in which aspects? can you please elaborate it more.

- Line 101-102 : To our knowledge, no prior empirical research in the smartphone sales industry has combined.................... ( For me it is kind of red-line How you know that there is no prior study? what if right now some one is looking to this matter? do you have access to all data sources? this is highly discouraged in research writing. You can write this: There is serious dearth of literature examining the relationship OR the review of literature shows a little research on ......

line 107-108: You mentioned " samples from different nations are also helpful in showing the results of

108 consumer perceptions of BGT in various national settings" Please explain why did you chose three countries to examine this? please give proper justification in context of your RQs. (US, France and Keniya) considering there is massive gap in culture, geo-political situation and economy.

Literature seems fine to me. but if you please show more connectivity of the theory to the research problem and RQs

Methods:

- Please explain overall research design? research approach?

- June 23rd 2022, to August 26th, 2022, Please explain you took this duration for the sample? is there any important consideration. if yes please write it.

- What was the study population? Respondents Profile? and more specifically Inclusive and Exclusive Criteria?

- Please elaborate how you conduct online data collection process considering three countries? do you have any links? or who was the gate keeper there?

- The online portals Facebook pages, WhatsApp groups, authenticity is in question always, can you write in more scientific way? It feels to me that it is just a survey of the opinions not the research data collection.

- Sampling technique is missing? Purposive? or do you have complete sampling frame?

It is good that this study is approved by ethical board. But my concern is how you ensure ethics in data collection process?

Results/Discussions

_ For me results are fine. but i request some changes in the discussion section.

_ Practical/Managerial Implications are missing. Please add.

Line 584-586: First, just three nations were examined in this study (Kenya,

585 France and the United States). Future research should include other developing and economically

Five hundred eighty-si developed nations to further understand the influence of BGT on customer purchase intention.

I don't know is this is a strength or limitation of the study. In these two sentences, you are contradicting your view. Please make it clear.

Language.

Please follow the academic writing style. The words like "just" are informal and preferred not used. On some occasions you wrotie planned theory behavior (It is theory of planned behavior)

I would advise to proofread the paper from professional editor.

I wish you all the best and look forward to see your revised manuscript.

Thanks

6. PLOS authors have the option to publish the peer review history of their article (what does this mean?). If published, this will include your full peer review and any attached files.

Reviewer #1: No

Reviewer #2: **Yes: **Dr. Ali Raza

---

## [Author Response · Author response to Decision Letter 0]

6 Dec 2022

Reviewer #1: Yes empirically I agree the claim on mediating role between BGT and PIT but how do you fit in the theory of planned behavior needs more elaboration. 

We sincerely appreciate your positive comments on mediating role between BGT and PIT. Regarding its fit in the theory of planned behavior, we have improved the paper's quality based on your comments in this revision as follows:

The literature mentioned above shows that considering the customer's perception of the purchase budget can influence the customer's perception of quality (PPQ), price (PPR), and benefit (PB). Moreover, it has been shown that these customer perceptions could be crucial precursors to customers' PIT.

Additionally, it has been highlighted that the customer's perception of purchase budget (BGT) incorporates both the perception of his planning obligations and choice expectations [9-10] e.i., his expectations in terms of price, quality, values, and benefit. Etc. TPB describes these perceived expectations as one of the foundations of customer intention [9-10]. Moreover, TPB stipulates that customer intention is the closer determinant of customer behaviour [68]. In other words, a customer's perception of his purchase budget (BGT) could determine his expectations perception, which could influence his intention and behaviour. Therefore, a customer's perception of the purchase budget can assist in attaining positive perceptions of product quality, price, and benefits that, in turn, influence customer purchase intention. Due to the possible effects of customer's perception of purchase budget on PPQ, PPR, and PB (H2), along with the effect of PPQ, PPR, and PB on customers' PIT (H3), customer's perception of purchase budget is predicted to have favorable indirect impacts on PIT through perceived quality, perceived price, and perceived benefit.

Why only Kenya, France and US? Why not compare other emerging countries with US and France? Is Kenya leading the way in cell phone usage? Please compare its facts and figures with other regional countries. 

Thanks for this question. As we have highlighted in the manuscript, We chose the USA, France, and Kenya for this study because: 

1) These countries are significant countries in smartphone penetration rates in North America, Europe, and Africa, respectively [75-76-77] and 2) these countries are considered to be among the major countries in purchasing smartphones through CBEC in North America, Europe, and Africa [78-79]. But also mainly because of the lack of data.

Yes, as we have already stated, we focused on Kenya not because Kenya is leading the way in cell phone usage worldwide but because it was among the countries leading the way in smartphone purchases through cross-border e-commerce transactions in Africa [73]. However, considering your comment, we have improved the article by comparing smartphone facts and figures of countries under study with other emerging countries. Figure.2 shows smartphone penetration rate in various nations based on data from [75-76-77]

The results are pretty similar for the three countries. Don't you think the purchasing power and culture may cause result differentials?

Thanks for your constructive comments. Yes, the results seem to be similar for the three countries. And we agree with you, as we have already stated that customer responses and perceptions of the purchase budget (BGT) might not be the same everywhere. Countries' purchasing power, tastes, and living conditions might generate differences [22–23]. Therefore, considering customer differences becomes crucial for e-sellers looking to succeed in the global online market [24]. In this context, the tiers of countries' economies may be utilized to explain the notable variances in buying power and living standards worldwide [25]. Regularly, consumers in developed nations show tendencies to purchase more expensive products than those in developing countries [26]. Customer answers might also differ based on product features [27]. Therefore, the perception of BGT could differ depending on customer preferences and quality expectations, which are frequently mirrored in their judgment process [7]. However, it should be noted that the transactions under investigation here are cross-border e-commerce transactions requiring a certain living standard (access to the internet connection, smartphone or laptop, Ect) to be carried out, especially in developing countries like Kenya. Therefore, customers purchasing through this market seem to have certain levels of purchasing power. That could explain why the result seems similar in the three countries.

There is confusion whether purchase budget or perceived purchase budget is under consideration. Title and hypothesis tell different story.

We apologize for the confusion created by our previous Title that could mislead readers to understanding the content. For convenience, we revised the Title to fit the content.

Please read and cite the following papers:

Thanks for the kind suggestion. We read the papers as you suggested and cited them in the revised manuscript:

Thus, a convenience sampling technique was chosen to carry out the study rather than a probability sample method [83]

For instance, [92] concluded that smartphone brands should focus on more sophisticated quality attributes in the smartphone design they propose to the market since, because of social media, customer requirements in terms of quality will increase in future.

Reviewer #2:

1. Introduction.

- Can you elaborate on the specific problem that triggers you to conduct this study? Though you have mentioned some literature gaps which are very fine but I am looking towards the specific practical issue esp in three chosen countries. 

Thanks for your helpful comments in revising the manuscript. For convenience, we added and revised the introduction as follows:

To summarize, this study mainly intends to answer the following questions:

RQ1: What is the effect of BGT on PIT in smartphones-buying on international online shopping platforms?

RQ2: What is the mediating role of PPQ, PPR, and PB in the relationship between BGT and PIT?

RQ3: Do the impact of the customers' perceived purchase budget on PPQ, PPR, and PB are the same across different settings or countries?

- In line 108: You mentioned "The research will advance knowledge 109 of BGT-PIT links in cross-national contexts"...how it advances and in which aspects? Can you please elaborate it more?

Thanks for your helpful questions. We elaborated the revised manuscript as follows: The research will advance knowledge of BGT-PIT links in cross-national contexts since minimal studies have investigated that relationship. Especially in such contexts. Additionally, almost no previous study has compared consumer perceptions of purchase budget, the suggested mediators, and PIT across various country settings. The current study's cross-country context also sets it apart from earlier studies on smartphone purchase intentions.

- Line 101-102: To our knowledge, no prior empirical research in the smartphone sales industry has combined.................... (For me it is kind of red-line How you know that there is no prior study? what if right now someone is looking to this matter? do you have access to all data sources? this is highly discouraged in research writing. You can write this: There is serious dearth of literature examining the relationship OR the review of literature shows a little research on......

Thanks for your constructive comments in revising the manuscript. We sincerely apologize for that kind of sentence which constitutes a red line in academic writing. For convenience, we revised that sentence as you have suggested as follows: There is a serious dearth of literature in the smartphone sales industry combining the mediators found in this study into a single model to account for how consumer perceptions of BGT affect customer purchase intentions. As a result, there is still a dearth of understanding of the BGT variable's various directions [7].

line 107-108: You mentioned "samples from different nations are also helpful in showing the results of 108 consumer perceptions of BGT in various national settings" Please explain why did you chose three countries to examine this? Please give proper justification in context of your RQs. (US, France and Kenya) considering there is massive gap in culture, geo-political situation and economy.

Thanks for your valuable comments. Considering your comment, we have improved the article to explain why we chose the three countries to examine the RQs as follows:

1) These countries are significant countries in smartphone penetration rates in North America, Europe, and Africa, respectively [75-76-77] and 2) these countries are considered to be among the major countries in purchasing smartphones through CBEC in North America, Europe, and Africa [78-79].

2) 

Literature seems fine to me. But if you please show more connectivity of the theory to the research problem and RQs

Thanks for your enlightening comment. For convenience, we added a Theoretical foundations part to explain more connectivity of the theory to the research questions and revised the manuscript as follows:

2.1 Theoretical foundations

In the case of smartphones, a CBEC transaction's success depends on the strategies put in place by e-sellers to convert subjective perceptions of buyers on the product (e.g. perception of quality, price, benefit, etc.) into buyer purchase intention and decision-making. The distance between buyer and seller requires e-sellers to develop strategies to convert buyers' subjective perceptions of products into purchase intentions and decisions. Also important is the alignment of the buying intention and decision-making with the perception of the budget for purchases ([29- 38].

Therefore, it is critical to comprehend how a customer's purchase budget, as well as the subjective perceptions of the buyer on the product, such as the quality, price, benefits, Etc., influence his purchase intention. In this regard, [29] claimed that using the theory of planned behavior, customer perceptions of their purchase budget can influence consumers' perception of the product. These perceptions, in turn, may impact the customer's buying intent [29-39-40-41]. In context, it has been suggested that TPB can be used to investigate the links between customers' perception of purchase budget and customer purchase intention, as well as the mediating role of buyers' perceptions [39]. From this theory, [39] concluded that customers' perception of purchase budget by influencing buyers' subjective perceptions of the product, e. i, quality, price, and benefit.., could lead to influence their purchase intention. Based on that theory lens, we argue that when customers find a match between their perception of the purchase budget and their subjective perceptions of the product they wish to purchase, this will reinforce their purchase intention, which is a result of their positive perceptions of the product. Our argument seems to be supported by [38]. They pointed out that customers' perceptions of the purchase budget could impact their perception of obtaining the values sought of the product they would like to buy, which in turn will reinforce their purchase intention.

Methods:

- Please explain overall research design? Research approach?

Thanks for your enlightening comment. For convenience, we revised the Research design and approach as you have suggested as follows:

A descriptive, exploratory, and empirical research design was used in this study. This study aims to identify the effect of customer perception of purchase budget on purchase intention through PPQ, PPR, and PB as mediators. Thus, a convenience sampling technique was chosen to carry out the study rather than a probability sample method [83]. A well-structured questionnaire was used to collect primary data for the study. The sample was collected from consumers who had recently purchased one or more smartphones through an international online shopping platform.

- June 23rd 2022, to August 26th, 2022, Please explain you took this duration for the sample? is there any important consideration. if yes please write it.

Thanks for your enlightening questions. We revised that part as follows:

The survey ran for about two months, from June 23rd 2022, to August 26th, 2022, through an online questionnaire distribution. We used two months to get more representative and accurate data. This duration includes data collection and data cleaning.

- What was the study population? Respondents Profile? And more specifically Inclusive and Exclusive Criteria? 

Thanks for your constructive questions. The study population sample consumers who had recently purchased one or more smartphones through an international online shopping platform. Minor customers (under 18) were not allowed to fill out the survey.

- Please elaborate how you conduct online data collection process considering three countries? do you have any links? or who was the gate keeper there? 

Thanks for your enlightening questions. We created an online survey through Google Forms to collect the study sample (https://forms.gle/pS4GHXACnjNSe55i6). We then shared the link with several online respondents (CBEC buyer groups) in France, Kenya, and the United States.

- The online portals Facebook pages, WhatsApp groups, authenticity is in question always, can you write in more scientific way? It feels to me that it is just a survey of the opinions not the research data collection. 

Thanks for your enlightening questions and suggestions. We revised that part as follows:

The method of distributing online questionnaires was chosen because the study framework concerns the international online shopping market. Through this virtual framework, CBEB customers interact with e-sellers and each other, discussing and sharing their shopping experiences [84]. Therefore, the response rate would be higher as these frameworks are convenient for contacting CBEC buyers as they are the study's target respondents [84].

- Sampling technique is missing? Purposive? or do you have complete sampling frame?

Thanks for your enlightening questions. We revised that part as follows:

Thus, a convenience sampling technique was chosen to carry out the study rather than a probability sample method [83]. A well-structured questionnaire was used to collect primary data for the study. The sample was collected from consumers who had recently purchased one or more smartphones through an international online shopping platform. The survey ran for about two months, from June 23rd 2022 to August 26th, 2022, through an online questionnaire distribution. We used two months to get more representative and accurate data.

It is good that this study is approved by ethical board. But my concern is how you ensure ethics in data collection process? 

Thanks for your positive comment. First, to ensure ethics in the data collection process, we created the google form in such a way that each response submitted was saved in the google spreadsheet drive related to our Google Forms. Next, we reviewed the responses and cleaned the data. For instance, answers with missing data were deleted. We did our best to ensure ethics in the data collection process.

Results/Discussions

_ For me results are fine. But i request some changes in the discussion section.

Thanks for your constructive comments in revising the manuscript and the discussion part of this article. As you requested, some changes have been made in the discussion section.

_ Practical/Managerial Implications are missing. Please add.

Thanks for your valuable suggestion. We added a Practical and Managerial Implications part as you have required as follows:

5.1. Managerial and Practical Implications

This study's results will inform smartphone brand managers and CBEC e-sellers of the significance of links between customer perceptions of their purchase budget and purchase intention. Therefore, this research has substantial practical implications for e-sellers and smartphone brands that sell their products on international online platforms. First, it will guide them in implementing their sales strategies to attract more customer attention by offering products in line with the perceptions of the values customers wish to find in the product and their perceptions of the purchase budget. That would necessarily improve the customers' purchase intentions. Second, e-sellers and smartphone brands thus must focus on smartphone offers that consider the standard of living. That would strengthen the links between the perception of the purchase budget and the perceptions of the critical values customers i. e., price, quality and benefits customers wish to find in the product. Doing so will strengthen customer purchase intention and improve the product's positive image in the customer's mind [89]. As [90] stated, buyers are generally more attracted to products when they perceive that the product could match their purchasing power. Their purchasing power supports their perception of the purchase budget. The perceived purchase budget appears as an essential factor which may be positively associated with the perceptions of the values that the customer is looking for in the product ([29-91] and positively associated with the customer's purchase intention, as demonstrated in this current study. Smartphone industries should focus on product offers that strengthen the links between the perception of values that customers wish to find in the product they purchase and the perception of purchase budget. That would allow e-sellers to offer customers products matching customer purchase budgets.

This research has identified that PPR and PPQ significantly mediate between BGT and PIT in smartphones purchased on CBEC platforms. Consequently, the smartphone industry must give more importance to PPR and PPQ in their product design strategies by offering products that match customer perception in terms of price, quality and purchase budget to succeed easily in that market. Scholars have already pointed out that it is essential to implement that kind of strategy to ensure success within the CBEC marketplace [7-80]. For instance, [92] concluded that smartphone brands should focus on more sophisticated quality attributes in the smartphone design they propose to the market since, because of social media, customer requirements in terms of quality will increase in future. In this direction, [91] showed that more than 90% of new products are unsuccessful in marketplaces due to quality and pricing attributes that do not match customer expectations and perceptions. Therefore, managers and smartphone manufacturers must establish a bridge between the pricing and quality expectations as well as perception, and the perception of the purchase budget, which will lead to a positive purchase intention.

Line 584-586: First, just three nations were examined in this study (Kenya, 585 France and the United States). Future research should include other developing and economically developed nations to further understand the influence of BGT on customer purchase intention. I don't know is this is a strength or limitation of the study. In these two sentences, you are contradicting your view. Please make it clear. 

Thanks for your constructive comments. For convenience, we revised these sentences to make them more straightforward as follows:

This study has some limitations. First, only three nations were examined in this study (Kenya, France, and the United States). Future research should include other nations with different economic backgrounds to further study the influence of BGT on customer purchase intention.

Language.

Please follow the academic writing style. The words like "just" are informal and preferred not used. On some occasions you wrote planned theory behavior (It is theory of planned behavior)

Thanks for your helpful comments in revising the manuscript. For convenience, we corrected that error. 

I would advise to proofread the paper from professional editor.

Thanks for the kind suggestion. We tried our best to check and correct the grammatical, spelling, and other common errors before submission.

The authors greatly appreciate your efforts in improving the paper. We sincerely thank you for your time and attention in reviewing our manuscript.

---

## [Decision Letter · Decision Letter 1]

12 Dec 2022

Perception of the purchase budget (BGT) and purchase intention in smartphone selling industry: A cross-country analysis

PONE-D-22-28268R1

Dear Dr. N'da,

We’re pleased to inform you that your manuscript has been judged scientifically suitable for publication and will be formally accepted for publication once it meets all outstanding technical requirements.

Kind regards,

Mingyue Fan, Ph.D.

Academic Editor

PLOS ONE

Additional Editor Comments (optional):

Reviewers' comments:

Reviewer's Responses to Questions

**Comments to the Author**

1. If the authors have adequately addressed your comments raised in a previous round of review and you feel that this manuscript is now acceptable for publication, you may indicate that here to bypass the “Comments to the Author” section, enter your conflict of interest statement in the “Confidential to Editor” section, and submit your "Accept" recommendation.

Reviewer #1: All comments have been addressed

Reviewer #2: All comments have been addressed

2. Is the manuscript technically sound, and do the data support the conclusions?

Reviewer #1: Yes

Reviewer #2: Yes

3. Has the statistical analysis been performed appropriately and rigorously? 

Reviewer #1: Yes

Reviewer #2: Yes

4. Have the authors made all data underlying the findings in their manuscript fully available?

Reviewer #1: Yes

Reviewer #2: Yes

5. Is the manuscript presented in an intelligible fashion and written in standard English?

Reviewer #1: Yes

Reviewer #2: Yes

6. Review Comments to the Author

Reviewer #1: (No Response)

Reviewer #2: (No Response)

7. PLOS authors have the option to publish the peer review history of their article (what does this mean?). If published, this will include your full peer review and any attached files.

Reviewer #1: No

Reviewer #2: **Yes: **Dr. Ali Raza

---

## [Editor Report · Acceptance letter]

22 Dec 2022

PONE-D-22-28268R1 

Perception of the purchase budget (BGT) and purchase intention in smartphone selling industry: A cross-country analysis 

Dear Dr. N'da:

I'm pleased to inform you that your manuscript has been deemed suitable for publication in PLOS ONE. Congratulations! Your manuscript is now with our production department. 

Kind regards, 

on behalf of

Dr. Mingyue Fan 

Academic Editor

PLOS ONE